# Real-Time Recurrent Learning using Trace Units in Reinforcement Learning

**Esraa Elelimy, Adam White**[*]**, Michael Bowling**[*]**, Martha White**[*]
University of Alberta, Alberta Machine Intelligence Institute (Amii)
[*]Canada CIFAR AI Chair
`elelimy,amw8,mbowling,whitem@ualberta.ca`

## Abstract

Recurrent Neural Networks (RNNs) are used to learn representations in partially observable environments. For agents that learn online and continually interact with the environment, it is desirable to train RNNs with real-time recurrent learning (RTRL); unfortunately, RTRL is prohibitively expensive for standard RNNs. A promising direction is to use linear recurrent architectures (LRUs), where dense recurrent weights are replaced with a complex-valued diagonal, making RTRL efficient. In this work, we build on these insights to provide a lightweight but effective approach for training RNNs in online RL. We introduce Recurrent Trace Units (RTUs), a small modification on LRUs that we nonetheless find to have significant performance benefits over LRUs when trained with RTRL. We find RTUs significantly outperform other recurrent architectures across several partially observable environments while using significantly less computation.[1]

## 1  Introduction

Agents, animals, and people perceive their surrounding environment through imperfect sensory observations. When the state of the environment is partially observable, agents construct and maintain their own state from the stream of observations. The constructed *agent state* summarizes past environment-agent interactions in a form that is useful to predict and control future interactions [41]. Recurrent Neural Networks (RNNs) provide a flexible architecture for constructing agent state [19, 21, 13, 6, 10].

While standard RNN architectures have been mainly supplanted by Transformers [44], in online reinforcement learning settings where the agent learns while interacting with the environment, RNNs remain a promising direction to pursue [17, 11]. There are two main issues that limit the use of self-attention mechanisms from Transformers in online learning. First, calculating the similarity between each pair of points results in a computational complexity that is a function of $k^2$, where $k$ is the sequence length. Moreover, calculating the similarity between all pairs ignores the temporal order of the data points, which limits the usefulness of self-attention when the data is temporally correlated [47]. Second, we need access to the whole sequence of observations before taking an action or updating the learnable parameters, which is impractical in continual learning. While recent works have reduced the complexity of transformers from quadratic in the sequence length to linear [15, 16, 37], the entire sequence length is still needed to train such architectures. Gated Transformer-XL attempts to overcome this issue by keeping a moving window of previous observations [35]. A window of past observations does not scale well to long sequences—the computation is quadratic in the sequence length—and a window is one particular fixed function of history. The simpler recursive form in RNNs, on the other hand, can learn a great variety of functions of history from the data and is well

---

[1]Code available at `https://github.com/esraaelelimy/rtus`

38th Conference on Neural Information Processing Systems (NeurIPS 2024).

suited for updating the state online from a sequential data stream and have been shown to outperform transformers in such settings [23].

A key open question is how to efficiently train RNNs in online RL. We can divide the literature into methods that approximate Real-Time Recurrent Learning (RTRL) and those that restrict the recurrent architecture. RTRL [46] exploits the recursive nature of the gradient for RNNs, carrying forward the needed gradient information instead of unrolling the recurrent dynamics back in time like Truncated Backpropagation Through Time (T-BPTT) [45]. RTRL avoids storing past data but is so computationally expensive and is intractable for even moderately sized networks. Several methods approximate the RTRL gradient update, including NoBackTrack [33], Unbiased Online Recurrent Optimization [43, 5], Sparse N-Step Approximation (SnAp) [28]. All of these methods produce a biased gradient estimate. Other works have tried to approximate an unbiased gradient estimate of BPTT specifically for the case of policy gradient updates in RL. However, the approximation resulted in a high variance due to added stochasticity to the policy [1].

Methods in the second category usually restrict the RNN architecture to a diagonal RNN [9, 30], including Columnar Networks [18], the element-wise LSTM [17], and Independently Recurrent Neural Networks (IndRNNs) [20]. The RTRL algorithm is computationally efficient for such architectures. However, this approach sacrifices representation power and can perform poorly [18]. Recent work suggests overcoming the poor performance of diagonal RNNs with a small modification: having a complex-valued recurrent state instead of restricting it to real values [34]. In fact, as we will show in section 3.1, there exists an equivalence between using a dense linear recurrent layer and a diagonal recurrent layer with complex values, indicating no loss of representational capacity. LRUs have been combined with RTRL [48], though only empirically explored for supervised learning datasets.

In this work, we extend the insights from LRUs into the online RL setting. Our primary contribution are our experiments showing that such a lightweight recurrent architecture can outperform standard approaches like Gated Recurrent Units (GRUs) [4] in RL, with significantly less computation. To obtain this result, we propose a small extension on LRUs, which we call Recurrent Trace Units (RTUs). RTUs incorporate nonlinearity into the recurrence and use a slightly different parameterization than LRUs, but one we find is more amenable to the use of RTRL in online RL than LRUs. We extend Proximal Policy Optimization (PPO) [40] to use RTRL, ablating the decision choices we propose. We provide an in-depth study in an animal-learning prediction benchmark, showing that RTUs scale better than GRUs with increasing compute and number of parameters and that RTUs outperform alternative diagonal recurrent architectures trained with RTRL. We then show across numerous control environments that RTUs have comparable or better performance, compared to GRUs and LRUs.

## 2  Background

We formalize the problem setting as a Partially Observable Markov Decision Process (POMDP). At each time step $t = 0, 1, 2, \ldots$, the agent perceives an observation $\mathbf{x}_t$, a limited view of the state $\mathbf{s}_t \in \mathcal{S}$, and takes an action $A_t \in \mathcal{A}(\mathbf{s}_t)$. Depending on the action taken, the agent finds itself in a new state $\mathbf{s}_{t+1} \in \mathcal{S}$, observes the corresponding observation $\mathbf{x}_{t+1}$ and a reward $R_{t+1} \in \mathbb{R}$. In the online control setting, the agent's goal is to maximize the discounted sum of the received rewards. It may also make predictions about its environment, such as future observations' outcomes.

For prediction and control in a partially observable environment, the agent should use the stream of observations to construct its *agent state*. The agent state summarizes information from the history of the agent-environment interactions that are useful for prediction and control [41]. We could use the whole history up to $t$, namely $(\mathbf{x}_0, A_1, R_1, \mathbf{x}_1, A_2, R_2, \ldots \mathbf{x}_t)$, as the agent state. Though the history preserves all the information, it is not feasible to use directly. We want the agent to have constant memory and computation per time step and storing the whole history causes the memory and the computation to grow with time. Instead, the agent needs to compress this history into a concise representation. We refer to the agent's internal representation of the history at time $t$ as its agent state or its hidden state $\mathbf{h}_t$. The agent constructs its current agent state $\mathbf{h_t} \in \mathbb{R}^n$ from its previous agent state $\mathbf{h}_{t-1} \in \mathbb{R}^n$ and the recent observation $\mathbf{x}_t \in \mathbb{R}^d$ using a state-update function $\mathbf{g} : \mathbb{R}^n \times \mathbb{R}^d \to \mathbb{R}^n$: $\mathbf{h}_t = \mathbf{g}(\mathbf{h}_{t-1}, \mathbf{x}_t)$.

One way to learn this state-update function $\mathbf{g}$ is with a recurrent neural network (RNN). A simple form is a linear recurrent layer, where $\mathbf{g}(\mathbf{h}_{t-1}, \mathbf{x}_t) = \mathbf{W}_x \mathbf{x}_t + \mathbf{W}_h \mathbf{h}_{t-1}$ for weight matrices $\mathbf{W}_x$ and $\mathbf{W}_h$. We can also add a nonlinear activation, such as ReLU.

In general, we will write
$$\mathbf{h}_t = \mathbf{g}(\mathbf{h}_{t-1}, \mathbf{x}_t, \boldsymbol{\psi}),$$
where $\boldsymbol{\psi}$ are the learnable parameters in the network. The agent maps the agent state $\mathbf{h}_t$ to an output $\hat{y}_t$ and then receives a loss $\mathcal{L}_t \doteq \mathcal{L}(\hat{y}_t, y_t)$ indicating how far the output is from a target $y_t$. The agent updates $\boldsymbol{\psi}$ to minimize this loss over time.

Two main gradient-based algorithms are widely used to train RNNs: Truncated Backpropagation Through Time (TBPTT) and Real-Time Recurrent Learning (RTRL). T-BPTT specifies a truncation length $T$, which controls the number of steps considered when calculating the gradient [45]. As a result, the computation and memory complexities of T-BPTT are functions of the truncation length. Learning with T-BPTT involves a trade-off between the network's ability to look further back in time and its compute and memory requirements. Picking a large $T$ can be expensive, or require us to limit the network size, but picking too small of a $T$ can cause the agent to miss important relationships and so result in poor performance.

Williams and Zipser (1989) introduced the Real-time Recurrent Learning algorithm (RTRL) as a learning algorithm for continual recurrent learning. Instead of unrolling the recurrent dynamics back in time, RTRL computes the gradient using the most recent observation, and the gradient is calculated and carried from the last step [46]. Assuming the network parameters have not changed, this recursive form gives the exact gradient and does not suffer from the truncation bias inherent to T-BPTT. We provide a more detailed background on the BPTT and RTRL in Appendix A. In reality, the agent updates its parameters frequently, so the gradient information saved from previous time steps is stale, i.e., calculated w.r.t old parameters; yet, under the assumption of small learning rates, RTRL is known to converge [46]. These properties make RTRL ideal for online learning, but unfortunately, there is a catch: its computational complexity is quartic, of fourth order, in the size of $\mathbf{h}_t$, which can be prohibitively expensive. For this reason, we pursue a restricted diagonal form in this work, for which RTRL is efficient and linear in $\mathbf{h}_t$.

# 3 Recurrent Trace Units

In this section, we introduce Recurrent Trace Units (RTUs). We start by revisiting why complex-valued diagonals represent dense recurrent layers, and why using real-valued diagonals is insufficient. We then introduce the specific form for RTUs that leverages this relationship. We then provide the RTRL update for RTUs, highlighting that it is simple to implement and linear in the hidden dimension. We finally contrast RTUs to LRUs and motivate why this small extension beyond LRUs is worthwhile.

## 3.1 Revisiting Complex-valued Diagonal Recurrence

Assume we have the recurrence relationship, with learnable parameters $\mathbf{W}_h \in \mathbb{R}^{n \times n}$ and $\mathbf{W}_x \in \mathbb{R}^{n \times d}$, $\mathbf{h}_t \doteq \mathbf{W}_h \mathbf{h}_{t-1} + \mathbf{W}_x \mathbf{u}(\mathbf{x}_t)$, where $\mathbf{u}$ can be any transformation of the inputs $\mathbf{x}_t$ before they are inputted into the recurrent layer. We can rewrite the square matrix $\mathbf{W}_h$ using an eigenvalue decomposition $\mathbf{W}_h = \mathbf{P} \boldsymbol{\Lambda} \mathbf{P}^{-1}$, where $\mathbf{P}$ contains the $n$ linearly independent eigenvectors and $\boldsymbol{\Lambda} \in \mathbb{C}^{n \times n}$ is a diagonal matrix with the corresponding eigenvalues. Then we have that

$$\mathbf{h}_t = \mathbf{P}(\boldsymbol{\Lambda} \mathbf{P}^{-1} \mathbf{h}_{t-1} + \mathbf{P}^{-1} \mathbf{W}_x \mathbf{u}(\mathbf{x}_t)) \implies \mathbf{P}^{-1} \mathbf{h}_t = \boldsymbol{\Lambda} \mathbf{P}^{-1} \mathbf{h}_{t-1} + \mathbf{P}^{-1} \mathbf{W}_x \mathbf{u}(\mathbf{x}_t)$$

By defining $\overline{\mathbf{h}}_t \doteq \mathbf{P}^{-1} \mathbf{h}_t \in \mathbb{C}^n$ and $\overline{\mathbf{W}}_x \doteq \mathbf{P}^{-1} \mathbf{W}_x \in \mathbb{C}^{n \times d}$, we get a new recurrence $\overline{\mathbf{h}}_t = \boldsymbol{\Lambda} \overline{\mathbf{h}}_{t-1} + \overline{\mathbf{W}}_x \mathbf{u}(\mathbf{x}_t)$.

We can see $\overline{\mathbf{h}}_t$ and $\mathbf{h}_t$ are representationally equivalent: they are linearly weighted for downstream predictions, and so the linear transformation on $\overline{\mathbf{h}}_t$ can fold into this downstream linear weighting. But it is more computationally efficient to use $\overline{\mathbf{h}}_t$ with a diagonal weight matrix $\boldsymbol{\Lambda}$, meaning each hidden unit only has one recurrent relation instead of n. LRUs precisely leverage this equivalence [34]. Specifically, they learn a complex-valued $\overline{\mathbf{h}}_t$, and use $\text{Re}(\overline{\mathbf{W}} \, \overline{\mathbf{h}}_t)$ as an input to an MLP for downstream nonlinearity.

Since we did not impose constraints on the matrix $\mathbf{W}_h$, other than being diagonalizable, the eigenvalues of $\mathbf{W}_h$ can be complex or real numbers. Previous diagonal RNNs such as eLSTM [17], Columnar networks [18], and IndRNN [20] use only real-valued diagonal matrices. Having only real-valued diagonals assumes that the matrix $\mathbf{W}_h$ is symmetric. We provide a small experiment in Appendix B.2 showing that this assumption does not hold even in the simplest setting and that complex eigenvalues do arise. We also investigate whether this result can be extended beyond linear recurrence, and largely obtain a negative theortical result (see Appendix B.1 and C ).

## 3.2 The RTU Parameterization

A complex number can be represented in three ways: $a + bi$ (the real representation), $r \exp(i\theta)$ (the exponential representation), and $r(\cos(\theta) + i \sin(\theta))$ (the cosine representation). Mathematically, these three representations are equivalent, but do they affect learning differently? Orvieto et al. [34] empirically showed that using the exponential representation resulted in a better-behaved loss function than the real representation on a simple task; we provide some discussion in Appendix D.1 further motivating why the real representation is less stable. We chose instead to pursue the cosine representation, because it allows us to represent the complex hidden vector as two real-valued vectors. The remainder of this section outlines RTUs, with and without nonlinearity in the recurrence.

Our goal is to learn a complex-valued diagonal matrix with weights $\lambda_k = r_k(\cos(\theta_k) + i \sin(\theta_k))$ on the diagonal, for $k = 1, \ldots, n$. Multiplying by a complex number is equivalent to multiplying by a 2x2 block matrix with a rescaling. We can use this rotational form to avoid explicitly using complex numbers, and instead use two real-values for each complex-valued hidden node. We write this real-valued matrix $\mathbf{\Lambda} \in \mathbb{R}^{2n \times 2n}$ as blocks of rotation matrices[2]

$$\mathbf{\Lambda} = \begin{bmatrix} \mathbf{c}_1 & & \\ & \cdots & \\ & & \mathbf{c}_n \end{bmatrix} \quad \text{where} \quad \mathbf{c}_k = \mathbf{r}_k \begin{bmatrix} \cos(\theta_k) & -\sin(\theta_k) \\ \sin(\theta_k) & \cos(\theta_k) \end{bmatrix}. \tag{1}$$

Each element of $\mathbf{h}_t = \mathbf{\Lambda}\mathbf{h}_{t-1} + \mathbf{W}_x \mathbf{x}_t \in \mathbb{R}^{2n}$ has two components $\mathbf{h}_t^{c_1}, \mathbf{h}_t^{c_2}$, updated recursively:

$$\mathbf{h}_t^{c_1} = \mathbf{r}\cos(\boldsymbol{\theta}) \odot \mathbf{h}_{t-1}^{c_1} - \mathbf{r}\sin(\boldsymbol{\theta}) \odot \mathbf{h}_{t-1}^{c_2} + \mathbf{W}_x^{c_1}\mathbf{x}_t,$$
$$\mathbf{h}_t^{c_2} = \mathbf{r}\cos(\boldsymbol{\theta}) \odot \mathbf{h}_{t-1}^{c_2} + \mathbf{r}\sin(\boldsymbol{\theta}) \odot \mathbf{h}_{t-1}^{c_1} + \mathbf{W}_x^{c_2} \mathbf{x}_t.$$

We finally combine the new recurrent states into one state $\mathbf{h}_t \doteq [\mathbf{f}(\mathbf{h}_t^{c_1}); \mathbf{f}(\mathbf{h}_t^{c_2})]$, potentially using a non-linearity $f$ after the recurrence.

We also adopt two parameterization choices made in LRUs that showed improved performance. The first is learning logarithmic representations of the parameters rather than learning them directly: instead of learning $\mathbf{r}$ and $\boldsymbol{\theta}$, the network learns $\boldsymbol{\nu}^{\log}$ and $\boldsymbol{\theta}^{\log}$, where $\mathbf{r} \doteq \exp(-\boldsymbol{\nu}), \boldsymbol{\nu} = \exp(\boldsymbol{\nu}^{\log})$, and $\boldsymbol{\theta}^{\log} \doteq \log(\boldsymbol{\theta})$. This re-parametrization restricts the $\mathbf{r}$ to be $\in (0, 1]$, required for stability. We found these modifications to improve stability of RTUs (see Appendix E). The second parameterization choice we adopt from LRUs is to multiply the input $(\mathbf{W}_x\mathbf{x}_t)_k$ by a normalization factor of $\gamma_k = (1 - r_k^2)^{1/2}$. Putting this all together, the final formulation of RTUs is:

$$\mathbf{h}_t^{c_1} = \mathbf{g}(\boldsymbol{\nu}^{\log}, \boldsymbol{\theta}^{\log}) \odot \mathbf{h}_{t-1}^{c_1} - \boldsymbol{\phi}(\boldsymbol{\nu}^{\log}, \boldsymbol{\theta}^{\log}) \odot \mathbf{h}_{t-1}^{c_2} + \boldsymbol{\gamma} \odot \mathbf{W}_x^{c_1}\mathbf{x}_t,$$
$$\mathbf{h}_t^{c_2} = \mathbf{g}(\boldsymbol{\nu}^{\log}, \boldsymbol{\theta}^{\log}) \odot \mathbf{h}_{t-1}^{c_2} + \boldsymbol{\phi}(\boldsymbol{\nu}^{\log}, \boldsymbol{\theta}^{\log}) \odot \mathbf{h}_{t-1}^{c_1} + \boldsymbol{\gamma} \odot \mathbf{W}_x^{c_2}\mathbf{x}_t, \tag{2}$$
$$\mathbf{h}_t = [\mathbf{f}(\mathbf{h}_t^{c_1}); \mathbf{f}(\mathbf{h}_t^{c_2})],$$

where $\boldsymbol{\gamma} \in \mathbb{R}^n$ is the vector composed of $\gamma_k = (1 - \exp(-\exp(\nu_k^{\log}))^2)^{1/2}$ and

$$g(\nu_k, \theta_k) \doteq \exp(-\exp(\nu_k^{\log})) \cos(\exp(\theta_k^{\log})),$$
$$\phi(\nu_k, \theta_k) \doteq \exp(-\exp(\nu_k^{\log})) \sin(\exp(\theta_k^{\log})). \tag{3}$$

Note that $\boldsymbol{\gamma}$ can be absorbed by $\mathbf{W}$, and so does not change representation capacity.

There are two ways to incorporate non-linearity into RTUs: inside the recurrence or after the recurrence. In the above, in Equation (2), the non-linearity is after the recurrence. These RTUs

---

[2]We assume the matrix $\mathbf{\Lambda}$ has only complex eigenvalues, as the network can easily turn a complex eigenvalue into a real one by setting the imaginary component to 0.

maintain the equivalence to a dense linear RNN, because the recurrence itself remains linear. We refer to this definition of RTUs as *Linear RTUs*, because the recurrence is linear, even though we have the ability to represent nonlinear functions by allowing for any nonlinear activation after the recurrence. We also evaluated a different variation of RTUs where the non-linearity is added to the recurrence directly. These *Nonlinear RTUs* are written as:

$$\mathbf{h}_t^{c_1} = \mathbf{f}(\mathbf{g}(\boldsymbol{\nu}^{\log}, \boldsymbol{\theta}^{\log}) \odot \mathbf{h}_{t-1}^{c_1} - \boldsymbol{\phi}(\boldsymbol{\nu}^{\log}, \boldsymbol{\theta}^{\log}) \odot \mathbf{h}_{t-1}^{c_2} + \boldsymbol{\gamma} \odot \mathbf{W}_x^{c_1} \mathbf{x}_t),$$
$$\mathbf{h}_t^{c_2} = \mathbf{f}(\mathbf{g}(\boldsymbol{\nu}^{\log}, \boldsymbol{\theta}^{\log}) \odot \mathbf{h}_{t-1}^{c_2} + \boldsymbol{\phi}(\boldsymbol{\nu}^{\log}, \boldsymbol{\theta}^{\log}) \odot \mathbf{h}_{t-1}^{c_1} + \boldsymbol{\gamma} \odot \mathbf{W}_x^{c_2} \mathbf{x}_t), \tag{4}$$
$$\mathbf{h}_t = [\mathbf{h}_t^{c_1}; \mathbf{h}_t^{c_2}].$$

Notice now $f$—a nonlinear activation like ReLU—is used in the update to $\mathbf{h}_t^{c_1}$ and $\mathbf{h}_t^{c_2}$, and the final $\mathbf{h}_t$ simply stacks these two components. Nonlinear RTUs lose the equivalence to dense RNNs, though in our experiments, we find they perform as well or better than Linear RTUs.

### 3.3 The RTRL Update for RTUs

This section shows the RTRL updates for RTUs with more in-depth derivations in Appendix E. To keep notation simpler, we write the updates as if we are directly updating $r$ and $\theta$; the updates for $\boldsymbol{\nu}^{\log}$ and $\boldsymbol{\theta}^{\log}$ are easily obtained then using the chain rule. The full derivation is in Appendix E.2.

Consider the partial derivative with respect to $r_1$ for the first RTU with input $\bar{x}_1 \doteq (\mathbf{W}_x^{c_1} \mathbf{x}_t)_1$:

$$h_{t,1}^{c_1} = r_1 \cos(\theta_1) h_{t-1,1}^{c_1} - r_1 \sin(\theta_1) h_{t-1,1}^{c_2} + \sqrt{(1 - r_1^2)} \bar{x}_1.$$

Then
$$\frac{\partial \mathcal{L}_t}{\partial r_1} = \frac{\partial \mathcal{L}_t}{\partial h_{t,1}^{c_1}} \frac{\partial h_{t,1}^{c_1}}{\partial r_1} + \frac{\partial \mathcal{L}_t}{\partial h_{t,1}^{c_2}} \frac{\partial h_{t,1}^{c_2}}{\partial r_1}.$$

Since $r_1$ only impacts the two units in the first RTU, and derivatives w.r.t. the remaining hidden units are zero. Therefore, we just need to keep track of the vector of partial derivatives for these two components, $\mathbf{e}_t^{r,c_1} \doteq \left[ \frac{\partial h_{t,1}^{c_1}}{\partial r_1}, \dots, \frac{\partial h_{t,n}^{c_1}}{\partial r_n} \right]$ and $\mathbf{e}_t^{r,c_2} \doteq \left[ \frac{\partial h_{t,1}^{c_2}}{\partial r_1}, \dots, \frac{\partial h_{t,n}^{c_2}}{\partial r_n} \right]$ with recursive formulas:

$$\mathbf{e}_t^{r,c_1} = \cos(\theta) \odot \mathbf{h}_{t-1}^{c_1} + \mathbf{r} \odot \cos(\theta) \odot \mathbf{e}_{t-1}^{r,c_1} - \sin(\theta) \odot \mathbf{h}_{t-1}^{c_2} - \mathbf{r} \odot \sin(\theta) \odot \mathbf{e}_{t-1}^{r,c_2} - \frac{\mathbf{r}}{\sqrt{1-\mathbf{r}^2}} \odot \mathbf{W}_x^{c_1} \mathbf{x}_t$$

$$\mathbf{e}_t^{r,c_2} = \cos(\theta) \odot \mathbf{h}_{t-1}^{c_2} + \mathbf{r} \odot \cos(\theta) \odot \mathbf{e}_{t-1}^{r,c_2} + \sin(\theta) \odot \mathbf{h}_{t-1}^{c_1} + \mathbf{r} \odot \sin(\theta) \odot \mathbf{e}_{t-1}^{r,c_1} - \frac{\mathbf{r}}{\sqrt{1-\mathbf{r}^2}} \odot \mathbf{W}_x^{c_2} \mathbf{x}_t$$

We can similarly derive such traces for $\theta$. The update to $\mathbf{r}$ involves first computing $\frac{\partial \mathcal{L}_t}{\partial \mathbf{h}_t^{c_1}}$, using backpropagation to compute gradients back from the output layer to the hidden layer; this step involves no gradients back-in-time. Then $\mathbf{r}$ is updated using the gradient $\frac{\partial \mathcal{L}_t}{\partial \mathbf{h}_t^{c_1}} \odot \mathbf{e}_t^{r,c_1} + \frac{\partial \mathcal{L}_t}{\partial \mathbf{h}_t^{c_2}} \odot \mathbf{e}_t^{r,c_2}$, which is linear in the size of $\mathbf{r} \in \mathbb{R}^n$, as the vectors $\mathbf{e}_t^{r,c_1}, \mathbf{e}_t^{r,c_2} \in \mathbb{R}^n$ can be updated with linear computation in the above recursion. This update is the RTRL update, with no approximation.

### 3.4 Contrasting to LRUs

RTUs are similar to LRUs, with two small differences. First, RTUs have real-valued hidden units, because the cosine representation is used instead of the exponential representation. Second, RTUs use nonlinear activations in the recurrence, making them no longer linear. Though again a minor difference, we find that incorporating nonlinearity in the recurrence can be beneficial. RTUs can be seen as a small generalization of LRUs, moving away from strict linearity—and thus motivating the name change—but nonetheless a generalization we find performs notably better in practice.

Let us now motivate the utility of moving to a cosine representation and real-valued traces. LRUs parameterize each hidden unit with $\lambda_k = r_k \exp(i\theta_k) = \exp(-\exp(\nu_k^{\log}) + i \exp(\theta_k^{\log}))$ and directly work with complex numbers. Consequently, the hidden layer cannot be directly used to predict real-values. It would be biased to take $\text{Re}(\bar{\mathbf{h}}_t)$ (see Appendix D.2), and instead an additional weight matrix $\overline{\mathbf{W}} \in \mathbb{C}^{n \times n}$ must be learned, to get $\text{Re}(\overline{\mathbf{W}} \bar{\mathbf{h}}_t)$. To understand why this works, assume that we took the original $\mathbf{h}_t$ from the dense NN, and handed it to an MLP. This would involve multiplying $\mathbf{W}\mathbf{h}_t$ for some $\mathbf{W}$. If we set $\overline{\mathbf{W}} = \mathbf{W}\mathbf{P}$, then $\overline{\mathbf{W}} \bar{\mathbf{h}}_t = \mathbf{W}\mathbf{P}\mathbf{P}^{-1}\mathbf{h}_t = \mathbf{W}\mathbf{h}_t$ and we did not introduce any bias. In fact, if $\overline{\mathbf{W}}$ is set this way, we do not need to take the real-valued part, because the output of $\overline{\mathbf{W}}\mathbf{h}_t$ is real-valued. Of course, learning does not force this equivalence—in fact this parameterization is more flexible than the original—and so it is necessary to take the real-part.

RTUs avoid some of these complications by explicitly writing the recurrence and updates with real-valued hidden states. Implicitly, the relationship between the two real-valued hidden vectors forces them to behave like complex numbers (as rotations), but all equations and learning stay in real-valued space. RTUs consequently avoid the need to post multiply by the matrix, removing a small number of learnable parameters, allowing the use of a nonlinear activation directly on the output, and allowing the hidden state to be immediately passed to a downstream MLP. We acknowledge that others may argue that working directly with complex numbers is preferable. The preference for real-valued hidden layers may simply be our own limitations, but we suspect much of the reinforcement learning community is similarly more comfortable to work in real-valued space. We found small choices in our implementation for LRU did not always behave as expected, partially due to how auto-differentiation is implemented in packages for complex numbers[3]. In the end, our goal is to make these simple recurrent traces easy to use, and providing updates with real numbers may remove some barriers.

## 4 Online Prediction Learning

In this section, we explore different architectural variants of RTUs and LRUs in a online prediction task and then move on to study the tradeoffs between computational resources and performance when using RTUs with RTRL compared to GRUs and LRUs with T-BPTT.

### 4.1 Ablation Study on Architectural Choices for RTUs and LRUs

In this first experiment, we investigate the impact of several architectural choices on the performance of RTUs and LRUs varying where nonlinearity is applied. We use a simple multi-step prediction task called *Trace conditioning* [39] inspired by experiments in animal learning. The agent's objective is to predict a signal—called the Unconditional Stimulus (US)—conditioned on an earlier signal—the Conditional Stimulus (CS). The prediction is formulated as a return, $G_t \doteq \sum_{k=0}^{\infty} \gamma^k \mathrm{US}_{t+k+1}$, where the agent's goal is to estimate the value function for this return. More details on this environment and experimental settings are in Appendix F. Figure 1 summarizes the results.

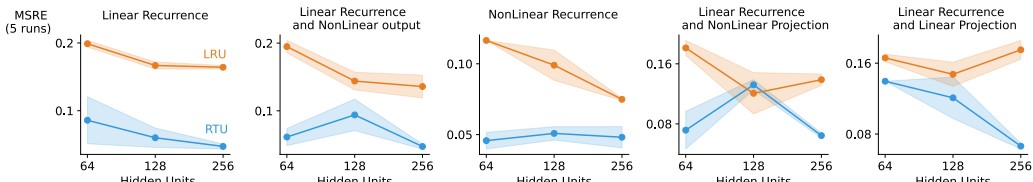

Figure 1: Ablation over different architectural choices for RTUs and LRUs. The RTU variants are blue, and the LRU variants are orange. In each subplot, we restrict both architectures in a particular way, reporting prediction error (MSRE) as a function of hidden state size. Across variations, RTUs are often better and, at worst, tie LRU. Here, both architectures were using RTRL.

### 4.2 Learning under resources constraints

In this section, we investigate the tradeoffs between computational resources and performance when using RTUs with RTRL compared to GRUs and LRUs with T-BPTT.

In the following experiments, all agents consist of a recurrent layer followed by a linear layer generating the prediction. We measure performance of the agents online by calculating the Mean Square Return Errors (MSRE), which is the mean squared error between the agent's prediction at time $t$ and $G_t$. In all the experiments, we used the Adam optimizer. We first ran each agent with different step sizes for 5 runs and 2 million steps. We then averaged the MSRE over the 2 million steps and selected each agent's best step size value. Finally, we ran the agents with the best step size value for 10 runs, which we report here. We also report all agents' step size sensitivity curves in Appendix F.3.

---

[3]Autodiff can give unexpected results when dealing with complex numbers. For example, see the discussion `https://github.com/google/jax/discussions/6817`.

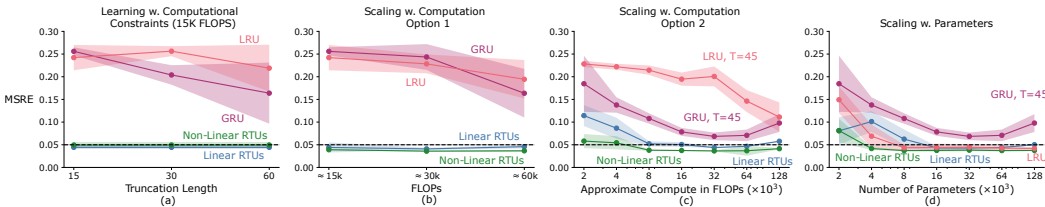

Figure 2: **Learning under resources constraints in Trace Conditioning.** Each of the four subplots shows how each algorithm's performance varies as a function of resources. (a) LRU and GRU with T-TBTT is not competitive with RTUs even as $T$ is increased while restricting the number of hidden units in LRU and GRU so that all algorithms use about the same computation per step. (b) If we allow GRU and LRU's computation to increase (fixed network size) while increasing $T$, the performance gap remains. (c) Fixing $T$ to a large value to solve the task, we can increase the number of parameters, holding the computation equal for all methods. (d) If we do not require compute to be equal across methods as we scale parameters, then the LRU can eventually match the error of RTU, but GRU cannot. The black dashed line represents the near perfect prediction performance.

**Learning under computational constraints:** We first investigate: *how well do different agents exploit the available computational resources?* We specified a fixed computational budget of 15000 FLOPs. Since RTUs are learning using RTRL and have a linear computational complexity, the computational budget only determines the number of hidden units in the architecture. For GRU and LRU, both the truncation length and the hidden dimension contribute to the budget. We tested several configurations of truncation lengths and parameters such that the overall computations fit the computational budget. Figure 2.a shows the results of this experiment. As we move along the horizontal axis, the number of parameters for GRU and LRU decreases as $T$ increases to fit the computational constraints. However, the RTU agents do not depend on $T$, so their performance and computation is constant.

**Scaling with computation:** The computational complexity of T-BPTT depends on the truncation length and the number of parameters in the neural network. Thus, the agent can use the additional resources in two ways: (1) Increasing the truncation length, and (2) Increasing the number of parameters. On the other hand, RTUs use all the computations to have more parameters.

Now, we move to our second question: *how well do different methods scale with increasing the computational budget?* We answer this question in two stages: Firstly, we study T-BPTT with increasing $T$ and a fixed number of parameters. For RTU, the computation increases by adding more parameters such that all corresponding points from GRU and RTU use the same amount of computation. Secondly, we fixed the truncation length for GRU to 45, which is more than the maximum distance between the CS and the US, and increased the computation by increasing the number of parameters for GRU. Again, for RTU, we increased the computation by increasing the number of parameters.

Figure of 2.b shows the first experiment's results. While GRU's performance improved as the truncation length increased, RTU outperformed GRU across all different computational budgets. Figure 2.c shows the results of the second experiment. The RTU agent's performance consistently improves as we increase the computation available. However, the performance improvement for the GRU agent is inconsistent. The inconsistency of GRU performance could be associated with the trade-off between the truncation length and the number of parameters.

**Scaling With Parameters:** Finally, we study the performance of RTU and GRU when given the same number of parameters and allow the GRU agent to use more computation. We fixed the truncation length for GRU to 45 as before and used the same number of parameters for both agents. Figure 2.d shows the results of this experiment. For RTU, we see the same consistent performance improvement as we increase the number of parameters. For GRU, the performance improvement is also consistent, though it degrades slightly towards the end. The RTU agent outperforms the GRU agent even though the GRU uses more computation.

We provide additional experiments comparing RTUs to two other approaches that use RTRL: online LRUs and a real-valued diagonal RNN in Appendix F.1.

# 5 Real-Time Recurrent Policy Gradient

This section first highlights some differences in using *linear RTRL* methods, i.e., RTRL with linear complexity, in incremental and batch settings. We then investigate different ways of integrating linear RTRL methods with policy gradient approaches, and we use PPO as a case study for this investigation. Finally, we compare the performance of RTRL methods with T-BPTT methods and other baselines.

## 5.1 Linear RTRL Methods in Incremental and Batch Settings

The benefits of linear RTRL methods over T-BPTT are more evident in the incremental rather than the batch setting. In the incremental learning setting, where the agent updates its parameters after each interaction step, linear RTRL methods have a constant computational complexity per update that depends only on the number of parameters. In contrast, T-BPTT methods have a complexity proportional to the truncation length T since T-BPTT methods require storing a sequence of past T activations to perform one gradient update. Figure 3 shows the time it takes to make one update with linear RTRL and T-BPTT given the same number of parameters. For T-BPTT, the time to make one update scales with the truncation length T, while for linear RTRL, it is constant.

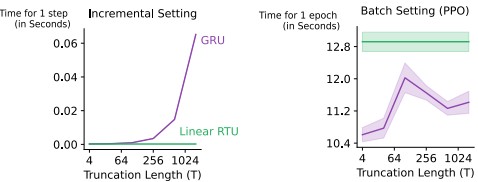

Figure 3: Contrasting runtime in incremental and batch settings. In the incremental setting, evaluated in the animal-learning prediction task, T-BPTT updates scale with truncation length, whereas linear RTRL is constant. With batch updates, evaluated in Ant-P with PPO, linear RTRL remains linear and T-BPTT is slightly more efficient.

The computational analysis for the batch setting is different than the incremental setting. In the batch setting, linear RTRL still have a constant cost per update and provide an untruncated yet stale gradient for all the samples. When using T-BPTT in the batch setting, there are two possibilities for the gradient updates. The first option, the typical strategy, is to divide the batch into non-overlapping sequences, each of length T, and perform T-BPTT on each sequence. In this case, the cost of one gradient update per sequence is a function of T, resulting in an effective update cost per sample independent of T. As a result, T-BPTT is computationally efficient in this case, albeit at the expense of a worse gradient estimate; in each sequence, only the last sample has a gradient estimate with T steps [25]. Figure 3 shows the time it takes to make one batch update with linear RTRL and T-BPTT given the same number of parameters. In this case, both methods use similar time per update. The second option is to divide the batch into overlapping sequences, where each gradient uses a sequence of T steps [25]. This approach increases the cost of updates per sample to be proportional to T, as in the incremental setting, with the benefit of better gradient estimates. However, all standard implementations of RL methods with T-BPTT use the computationally efficient option [38, 14, 24].

**Integrating Linear RTRL Methods with PPO** When performing batch updates, as with PPO, the RTRL gradients used to update the recurrent parameters will be stale, as they were calculated during the interaction with the environment w.r.t old policy and value parameters. One solution to mitigate the gradient staleness is to go through the whole trajectory after each epoch update and re-compute the gradient traces. However, this can be computationally expensive. In Appendix G, Algorithm 1, we provide the pseudocode for integrating RTRL methods with PPO with optional steps for re-running the network to update the RTRL gradient traces, the value targets, and the advantage estimates. We also performed an ablation study to investigate the effect of the gradient staleness in RTRL when combined with PPO, Appendix G. The results from the ablation study show that using a stale gradient results in better performance with RTUs and suggests that the staleness might help PPO maintain the trust region.

# 6 Experiments in Memory-Based Control

In this section, we evaluate the memory capabilities of RTUs when solving challenging RL control problems. We divide the problems in this section based on the source of partial observability. 1) Missing sensory data, where we mask out parts of the agent's observation. The agent must

accumulate and integrate the sensory observations over time to account for the missing information. 2) Remembering important cues, where the agent must remember an essential cue about the environment that happened many steps in advance.

**Integrating Sensory Observations:** We use the standard Mujoco POMDP benchmark widely used in prior work for evaluating memory-based RL agents [31, 12, 27, 32]. The benchmark consists of several challenging tasks where the agent controls a multi-joint dynamical body while only observing the joints' positional (Mujoco-P) or velocity information (Mujoco-V). To increase experiment throughput, we use the Jax implementation of Mujoco from the Brax library [7] and implemented wrappers to mask either the velocity (Mujoco-P) or positional information (Mujoco-V).

We evaluated our Linear and Non-linear RTUs against GRU, LRU, and Online LRU. All agents use PPO [40] as the control algorithm, and the difference between the agents is the recurrent component. For all agents, we fixed the number of parameters for the recurrent part to be $\sim 24k$. We tuned the learning rate for all agents in all environments and selected the best learning rate for each agent per environment. We also included a GPT2-transformer baseline. We followed the implementation details in previous work [32], and used a GPT2 variant with 200k parameters. We provide the results for GPT2 in Appendix H.

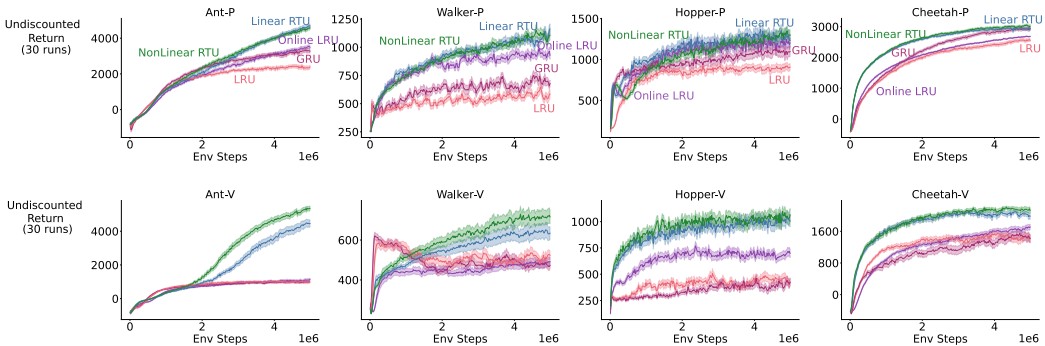

Figure 4: Learning curves on the Mujoco POMDP benchmark. Environments with -P mean that velocity components are occluded from the observations, while -V means that the positions and angles are occluded. All architectures have the same number of recurrent parameters ( 24k parameter). For each architecture, we show the performance of its best-tuned variant.

When given the same number of parameters, RTU agents outperform other baselines in all environments in Figure 4. Furthermore, we show in Appendix H that even when increasing the truncation length of both GRU and LRU agents to use significantly longer history, they outperform RTUs in only one task. Of particular note is again that RTUs outperform online LRUs, highlighting again that our simple modifications have a large impact on performance in this online RL setting.

**Remembering Important Cues:**
Next, we test the agents' ability to remember essential environmental cues. We use several tasks from the POPGym benchmark [29] in addition to the Reacher POMDP task, a modified version of Mujoco Reacher where the agent observes the target position only at the beginning of the episode.

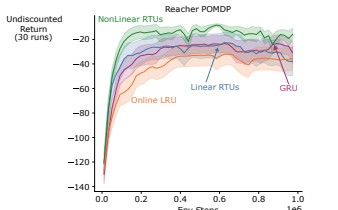

Figure 5: Reacher, 30 runs with standard errors.

The POPGym tasks we consider along with the Reacher POMDP are all long-term memory tasks [32] as the agent must remember and carry the information for a long time.

Figure 5 summarizes the results for the reacher POMDP task and the POPGym results can be found in Figure 6. In both cases, we can see that RTUs outperform the other approaches. Non-linear RTUs achieve a better performance than linear RTUs in reacher POMDP, and both achieve a better performance in all tasks than online LRUs. In Reacher POMDP, GRU was able to achieve a similar performance to that of linear RTUs.

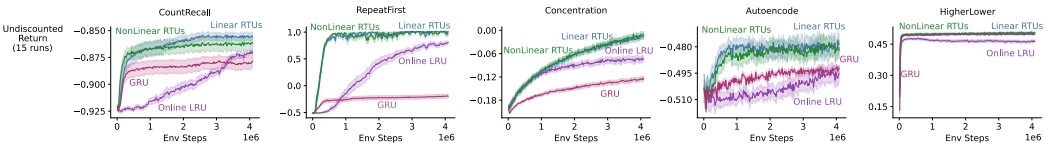

Figure 6: Results across several tasks from the POPGym benchmark.

# 7 Conclusion and Limitations

In this work, we investigated using complex-valued diagonal RNNs for online RL. We built on LRUs, to provide a small modification (RTUs) that we found performed significantly better in online RL across various partially observable prediction and control settings. We also found RTUs performed better than the more computationally intensive GRUs. Overall, RTUs are a promising, lightweight approach to online learning in partially observable RL environments.

A primary limitation of RTUs is the extension to multilayer recurrence. This limitation is inherent to all RTRL approaches; with multilayers, we need to save the gradient traces of the hidden state w.r.t the weights from all the preceding layers [17]. Previous work [17, 48] showed that using stop gradient operations between the layers and not tracing the time dependencies across layers is a viable choice. However, we need a more principled approach for tracing the gradient across layers.

One advantage of the linearity restriction in LRUs is that it allows the use of parallel scans for training [26]. However, recent works have shown the possibility of employing parallel scans to non-linear RNNs [8, 22]. A future direction is to investigate the use of parallel scans for training RTUs.

# 8 Acknowledgments

We would like to thank Nicolas Zucchet for advice about the online LRU implementation, and Subhojeet Pramanik for many discussions on transformers and RNNs. We would like to thank NSERC, CIFAR and Amii for research funding and the Digital Research Alliance of Canada for the computational resources.

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

# A Background on BackPropagation Through Time and Real-Time Recurrent Learning

This section provides a brief background on BackPropagation Through Time (BPTT) and Real-Time Recurrent Learning (RTRL) algorithms.

## A.1 BackPropagation Through Time

BPTT calculates the gradient, $\nabla_{\boldsymbol{\psi}}\mathcal{L}$, by unfolding the recurrent dynamics through time and incorporating the impact of the parameters on the loss from all observed time steps. Formally, we can write $\nabla_{\boldsymbol{\psi}}\mathcal{L}$ as:

$$\nabla_{\boldsymbol{\psi}}\mathcal{L} = \frac{1}{t}\sum_{i=0}^{t-1}\nabla_{\boldsymbol{\psi}}\mathcal{L}_i. \tag{5}$$

Applying the chain rule, we re-write Eq.5 as:

$$\begin{aligned}\nabla_{\boldsymbol{\psi}}\mathcal{L} &= \frac{1}{t}\sum_{i=0}^{t-1}\nabla_{\boldsymbol{\psi}}\mathcal{L}_i \\ &= \frac{1}{t}\sum_{i=0}^{t-1}\frac{\partial\mathcal{L}_i}{\partial\mathbf{h}_i}\frac{\partial\mathbf{h}_i}{\partial\boldsymbol{\psi}}.\end{aligned} \tag{6}$$

When calculating $\frac{\partial\mathbf{h}_i}{\partial\boldsymbol{\psi}}$, we need to consider the effect of $\boldsymbol{\psi}$ from all the time steps. To illustrate this effect, consider unrolling the last 2 steps of the RNN dynamics:

$$\begin{aligned}\mathbf{h}_t &= \mathbf{f}(\mathbf{h}_{t-1}, \mathbf{x}_t, \boldsymbol{\psi}) \\ &\quad\text{Re-write } \mathbf{h}_{t-1} \text{ as } \mathbf{f}(\mathbf{h}_{t-2}, \mathbf{x}_{t-1}, \boldsymbol{\psi}) \\ &= \mathbf{f}(\mathbf{f}(\mathbf{h}_{t-2}, \mathbf{x}_{t-1}, \boldsymbol{\psi}), \mathbf{x}_t, \boldsymbol{\psi}) \\ &\quad\text{Re-write } \mathbf{h}_{t-2} \text{ as } \mathbf{f}(\mathbf{h}_{t-3}, \mathbf{x}_{t-2}, \boldsymbol{\psi}) \\ &= \mathbf{f}(\mathbf{f}(\mathbf{f}(\mathbf{h}_{t-3}, \mathbf{x}_{t-2}, \boldsymbol{\psi}), \mathbf{x}_{t-1}, \boldsymbol{\psi}), \mathbf{x}_t, \boldsymbol{\psi}).\end{aligned} \tag{7}$$

Equation 7 shows that the network parameters $\boldsymbol{\psi}$ affect the construction of the recurrent state $\mathbf{h}_t$ through two pathways: a direct pathway, i.e., using $\boldsymbol{\psi}$ to evaluate $\mathbf{f}(\mathbf{h}_{t-1}, \mathbf{x}_t, \boldsymbol{\psi})$, and an implicit pathway, i.e., $\boldsymbol{\psi}$ affected constructing all previous recurrent states, $\mathbf{h}_{t-1}, \ldots, \mathbf{h}_1$, and all those recurrent states affected $\mathbf{h}_t$ construction. Thus, to calculate $\frac{\partial\mathbf{h}_t}{\partial\boldsymbol{\psi}}$, we need to consider those two pathways:

$$\frac{\partial\mathbf{h}_t}{\partial\boldsymbol{\psi}} = \frac{\partial\mathbf{f}(\mathbf{h}_{t-1}, \mathbf{x}_t, \boldsymbol{\psi})}{\partial\boldsymbol{\psi}} + \frac{\partial\mathbf{f}(\mathbf{h}_{t-1}, \mathbf{x}_t, \boldsymbol{\psi})}{\partial\mathbf{h}_{t-1}}\frac{\partial\mathbf{h}_{t-1}}{\partial\boldsymbol{\psi}}. \tag{8}$$

Once again, we need to consider the two pathways when evaluating $\frac{\partial\mathbf{h}_{t-1}}{\partial\boldsymbol{\psi}}$ in 8. For simplicity, let $\mathbf{J}_t \doteq \frac{\partial\mathbf{h}_t}{\partial\boldsymbol{\psi}}$, $\mathbf{B}_t = \frac{\partial\mathbf{f}(\mathbf{h}_{t-1}, \mathbf{x}_t, \boldsymbol{\psi})}{\partial\boldsymbol{\psi}}$, $\mathbf{C}_t = \frac{\partial\mathbf{f}(\mathbf{h}_{t-1}, \mathbf{x}_t, \boldsymbol{\psi})}{\partial\mathbf{h}_{t-1}}$, and re-write 8:

$$\begin{aligned}\mathbf{J}_t &= \mathbf{B}_t + \mathbf{C}_t\mathbf{J}_{t-1} \\ &= \mathbf{B}_t + \mathbf{C}_t\left(\mathbf{B}_{t-1} + \mathbf{C}_{t-1}\mathbf{J}_{t-2}\right) \qquad \text{Unrolling } \mathbf{J}_{t-1} \\ &= \mathbf{B}_t + \mathbf{C}_t\mathbf{B}_{t-1} + \mathbf{C}_t\mathbf{C}_{t-1}\mathbf{J}_{t-2} \\ &= \mathbf{B}_t + \mathbf{C}_t\mathbf{B}_{t-1} + \mathbf{C}_t\mathbf{C}_{t-1}\left(\mathbf{B}_{t-2} + \mathbf{C}_{t-2}\mathbf{J}_{t-3}\right) \qquad \text{Unrolling } \mathbf{J}_{t-2} \\ &= \mathbf{B}_t + \mathbf{C}_t\mathbf{B}_{t-1} + \mathbf{C}_t\mathbf{C}_{t-1}\mathbf{B}_{t-2} + \mathbf{C}_t\mathbf{C}_{t-1}\mathbf{C}_{t-2}\mathbf{J}_{t-3} \\ &= \mathbf{B}_t + \mathbf{C}_t\mathbf{B}_{t-1} + \mathbf{C}_t\mathbf{C}_{t-1}\mathbf{B}_{t-2} + \cdots + \mathbf{C}_t\mathbf{C}_{t-1}\mathbf{C}_{t-2}\ldots\mathbf{C}_2\mathbf{B}_1 + \mathbf{C}_t\mathbf{C}_{t-1}\mathbf{C}_{t-2}\ldots\mathbf{C}_1\mathbf{J}_0 \qquad \text{Keep unrolling} \\ &= \sum_{k=1}^{t}\left(\prod_{i=k+1}^{t}\mathbf{C}_i\right)\mathbf{B}_k + \left(\prod_{i=1}^{t}\mathbf{C}_i\right)\mathbf{J}_0.\end{aligned}$$

$$\tag{9}$$

Writing $\frac{\partial \mathbf{h}_t}{\partial \boldsymbol{\psi}}$ using the results from 9:

$$\frac{\partial \mathbf{h}_t}{\partial \boldsymbol{\psi}} = \frac{\partial \mathbf{f}(\mathbf{h}_{t-1}, \mathbf{x}_t, \boldsymbol{\psi})}{\partial \boldsymbol{\psi}} + \frac{\partial \mathbf{f}(\mathbf{h}_{t-1}, \mathbf{x}_t, \boldsymbol{\psi})}{\partial \mathbf{h}_{t-1}} \frac{\partial \mathbf{h}_{t-1}}{\partial \boldsymbol{\psi}}$$

$$= \sum_{k=1}^{t} \left( \prod_{i=k+1}^{t} \frac{\partial \mathbf{f}(\mathbf{h}_{i-1}, \mathbf{x}_i, \boldsymbol{\psi})}{\partial \mathbf{h}_{i-1}} \right) \frac{\partial \mathbf{f}(\mathbf{h}_{k-1}, \mathbf{x}_k, \boldsymbol{\psi})}{\partial \boldsymbol{\psi}} + \left( \prod_{i=1}^{t} \frac{\partial \mathbf{f}(\mathbf{h}_{i-1}, \mathbf{x}_i, \boldsymbol{\psi})}{\partial \mathbf{h}_{i-1}} \right) \frac{\partial \mathbf{h}_0}{\partial \boldsymbol{\psi}}. \tag{10}$$

According to Eq. 10, the agent needs to store all the previous inputs to calculate $\frac{\partial \mathbf{h}_t}{\partial \boldsymbol{\psi}}$ which is impractical; the computation and memory complexity will be increasing with $t$.

### A.1.1 Truncated-BackPropagation Through Time

Williams and Peng (1990) introduced Truncated-BackPropagation Through Time (T-BPTT) which solves the issue of increasing memory and computational complexities of BPTT. In T-BPTT, we specify a truncation length $T$, which controls the number of steps considered when calculating the gradient in 10. We now write the truncated version of 9 which takes into consideration the gradient from the last $T$ steps only:

$$\mathbf{J}_t = \sum_{k=t-T}^{t} \left( \prod_{i=k+1}^{t} \mathbf{C}_i \right) \mathbf{B}_k \tag{11}$$

Using results from 11, we then write the approximated gradient of the loss w.r.t the learnable parameters:

$$\nabla_{\boldsymbol{\psi}} \mathcal{L} = \sum_{j=t-T}^{t} \frac{\partial \mathcal{L}_j}{\partial \mathbf{h}_j} \frac{\partial \mathbf{h}_j}{\partial \boldsymbol{\psi}}$$

$$= \sum_{j=t-T}^{t} \frac{\partial \mathcal{L}_j}{\partial \mathbf{h}_j} \sum_{k=j-T}^{j} \left( \prod_{i=k+1}^{t} \frac{\partial \mathbf{f}(\mathbf{h}_{i-1}, \mathbf{x}_i, \boldsymbol{\psi})}{\partial \mathbf{h}_{i-1}} \right) \frac{\partial \mathbf{f}(\mathbf{h}_{k-1}, \mathbf{x}_k, \boldsymbol{\psi})}{\partial \boldsymbol{\psi}} + \left( \prod_{i=1}^{t} \frac{\partial \mathbf{f}(\mathbf{h}_{i-1}, \mathbf{x}_i, \boldsymbol{\psi})}{\partial \mathbf{h}_{i-1}} \right) \frac{\partial \mathbf{h}_0}{\partial \boldsymbol{\psi}} \tag{12}$$

### A.2 Real-Time Recurrent Learning

Williams and Zipser (1989) introduced the Real-time Recurrent Learning algorithm (RTRL) as a learning algorithm for continual recurrent learning. RTRL employs the recurrent formulation of the gradient in 8; instead of unrolling $\frac{\partial \mathbf{h}_{t-1}}{\partial \boldsymbol{\psi}}$ further back in time, RTRL saves its calculated value from the previous time step and use it later when needed. It is worth emphasizing that after the agent updates its parameters, the gradient information saved from previous time steps would be stale, i.e., calculated w.r.t old parameters, however, under the assumption of small learning rates, RTRL is known to converge. The gradient formulation of RTRL can be written as:

$$\nabla_{\boldsymbol{\psi}} \mathcal{L} = \sum_{i=0}^{t} \frac{\partial \mathcal{L}_i}{\partial \mathbf{h}_i} \frac{\partial \mathbf{h}_i}{\partial \boldsymbol{\psi}}$$

$$= \sum_{i=0}^{t} \frac{\partial \mathcal{L}_i}{\partial \mathbf{h}_i}$$

$$\left( \frac{\partial \mathbf{f}(\mathbf{h}_{i-1}, \mathbf{x}_i, \boldsymbol{\psi})}{\partial \boldsymbol{\psi}} + \frac{\partial \mathbf{f}(\mathbf{h}_{i-1}, \mathbf{x}_i, \boldsymbol{\psi})}{\partial \mathbf{h}_{i-1}} \frac{\partial \mathbf{h}_{i-1}}{\partial \boldsymbol{\psi}} \right) \tag{13}$$

## B  More Details on Representability with Complex-valued Diagonal Recurrence

This section explains why we need complex-valued diagonals to represent dense recurrent layers. We first show when it is equivalent to use complex-valued diagonal and a dense recurrent layer. We highlight that using a real-valued diagonal is like restricting the weights to be symmetric—because

the (complex) diagonal corresponds to the eigenvalues of the weight matrix—which can severely limit representability. We provide a small experiment to show that complex eigenvalues naturally arise when training both a dense linear and nonlinear RNN, further motivating the utility of moving towards complex-valued diagonals.

## B.1 Representability with Complex-valued Diagonals

Let us first consider when we can perfectly represent a dense, linear recurrent layer with a complex-valued diagonal recurrent layer. Assume we have the recurrence relationship, with learnable parameters $\mathbf{W}_h \in \mathbb{R}^{n \times n}$ and $\mathbf{W}_x \in \mathbb{R}^{n \times d}$,

$$\mathbf{h}_t \doteq f(\mathbf{W}_h \mathbf{h}_{t-1} + \mathbf{W}_x \mathbf{u}(\mathbf{x}_t)) \tag{14}$$

where $f$ is a potentially nonlinear function that inputs a vector and outputs the same-sized vector and $\mathbf{u}$ can be any transformation of the inputs $\mathbf{x}_t$ before they are inputted into the recurrent layer. The following equivalence result is straightforward but worthwhile formalizing.

**Proposition B.1.** *Assume $f \circ \mathbf{P} = \mathbf{P} \circ f$ for any full rank, potentially complex-valued $\mathbf{P} \in \mathbb{C}^{n \times n}$ with unit-length column vectors. Then given any $\mathbf{W}_h$ and $\mathbf{W}_x$ for Equation (14), there is a corresponding complex-valued diagonal weight matrix $\mathbf{\Lambda} \in \mathbb{C}^{n \times n}$ and $\overline{\mathbf{W}}_x \in \mathbb{C}^{n \times d}$*

$$\overline{\mathbf{h}}_t = f(\mathbf{\Lambda} \overline{\mathbf{h}}_{t-1} + \overline{\mathbf{W}}_x \mathbf{u}(\mathbf{x}_t)). \tag{15}$$

*where $\overline{\mathbf{h}}_t \in \mathbb{C}^n$ is a linear transformation of $\mathbf{h}_t \in \mathbb{R}^n$.*

*Proof.* We can rewrite the square matrix $\mathbf{W}_h$ using an eigenvalue decomposition $\mathbf{W}_h = \mathbf{P} \, \mathbf{\Lambda} \, \mathbf{P}^{-1}$, where $\mathbf{P}$ contains the $n$ linearly independent eigenvectors and $\mathbf{\Lambda}$ is a diagonal matrix with the corresponding eigenvalues. Then, we can re-write (14) as:

$$\begin{aligned} \mathbf{h}_t &= f(\mathbf{P} \, \mathbf{\Lambda} \, \mathbf{P}^{-1} \, \mathbf{h}_{t-1} \, + \, \mathbf{P}\mathbf{P}^{-1}\mathbf{W}_x \, \mathbf{u}(\mathbf{x}_t)) \\ &= \mathbf{P} f(\mathbf{\Lambda} \, \mathbf{P}^{-1} \, \mathbf{h}_{t-1} \, + \, \mathbf{P}^{-1}\mathbf{W}_x \, \mathbf{u}(\mathbf{x}_t)) \\ \mathbf{P}^{-1}\mathbf{h}_t &= f(\mathbf{\Lambda}\mathbf{P}^{-1} \, \mathbf{h}_{t-1} \, + \, \mathbf{P}^{-1} \, \mathbf{W}_x \, \mathbf{u}(\mathbf{x}_t)) \end{aligned} \tag{16}$$

where $\mathbf{P}$ came outside of $f$ under our assumption that it commutes with such matrices of eigenvectors. By defining $\overline{\mathbf{h}}_t \doteq \mathbf{P}^{-1} \, \mathbf{h}_t$ and $\overline{\mathbf{W}}_x \doteq \mathbf{P}^{-1}\mathbf{W}_x$, we get Eq. (15). □

We can see $\overline{\mathbf{h}}_t$ and $\mathbf{h}_t$ are representationally equivalent: they are linearly weighted for downstream predictions, and so the linear transformation on $\overline{\mathbf{h}}_t$ can fold into this downstream linear weighting. But it is more computationally efficient to use $\overline{\mathbf{h}}_t$ with a diagonal weight matrix $\mathbf{\Lambda}$, meaning each hidden unit only has one recurrent relation instead of n.

Since we did not impose constraints on the matrix $\mathbf{W}_h$, other than being diagonalizable, the eigenvalues of $\mathbf{W}_h$ can be complex or real numbers. Previous diagonal RNNs such as eLSTM [17], Columnar networks [18], and IndRNN [20] use only real-valued diagonal matrices. Having only real-valued diagonals implicitly assumes that the matrix $\mathbf{W}_h$ is a symmetric matrix. [34] suggested using complex-valued diagonal matrices for better performance.

The above equivalence has only been used without any activation, namely in linear-recurrent units (LRUs) [34]. A natural question is if only the identity $f(\mathbf{x}) = \mathbf{x}$ (and linear functions) satisfy this property of commuting with eigenvector matrices. Intuitively, this seems like the only option, as imagining a nonlinearity that commutes is hard. Surprisingly, for the more restricted case of symmetric $\mathbf{W}_h$, we can show a slightly more general class of activations can be used, proving an if-and-only-if relationship (see Appendix C). However, even for this restricted setting, this generalized class is limited and such activations unlikely to be preferable to a linear recurrence. We see this as a negative result, that suggests this equivalence only holds for the linear setting.

## B.2 Complex Eigenvalues in Vanilla RNNs

We empirically investigate whether complex eigenvalues appear when training dense RNNs in a simple task. The goal is to show that the weight matrix, $\mathbf{W}_h$, is not a symmetric matrix, even in the simplest tasks. Hence, having only real-valued diagonals is too restrictive.

We used the Three State POMDP [42], depicted in Figure 7, for this experiment. In this task, the agent needs to remember one cue from the previous time-step ago, to make a prediction about the next time-step. The MDP has three states, $s_1$, $s_2$, and $s_3$, and no actions. If the agent is in either $s_1$ or $s_2$, it transitions to any of the three states with equal probability. However, if the agent is in $s_3$, it transitions to the state preceded by $s_3$. A sequence of observations would look like $1, 3, 1, 2, 2, 3, 2, \cdots$, and we ask the agent to predict the next observation.

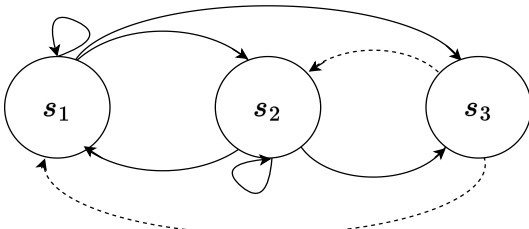

Figure 7: Illustration of the Three State MDP. We used dashed lines for the transitions starting in $s_3$ to make them more visible.

We trained a vanilla RNN with 3 hidden states with T-BPTT with truncation length 2, which is a sufficient history length in this problem to predict the next observation. Since we have 3 hidden states, the matrix $\mathbf{W}_h$ is $\in \mathbb{R}^{3\times 3}$ and could have at most 2 complex eigenvalues.

We measured the performance in terms of the percentage of correct predictions made in $S_3$. We recorded the number of complex eigenvalues of $\mathbf{W}_h$ after each parameter update, shown in Figure 8. This agent reaches $100\%$ accuracy in this problem relatively quickly. We can also see that the agent oscillates between having two complex eigenvalues and zero eigenvalues. The average number of complex eigenvalues across 30 run is above 1.5, which means that on more than $\frac{3}{4}$ of the steps, the RNN has two complex eigenvalues. The primary point is that we see complex eigenvalues appear frequently.

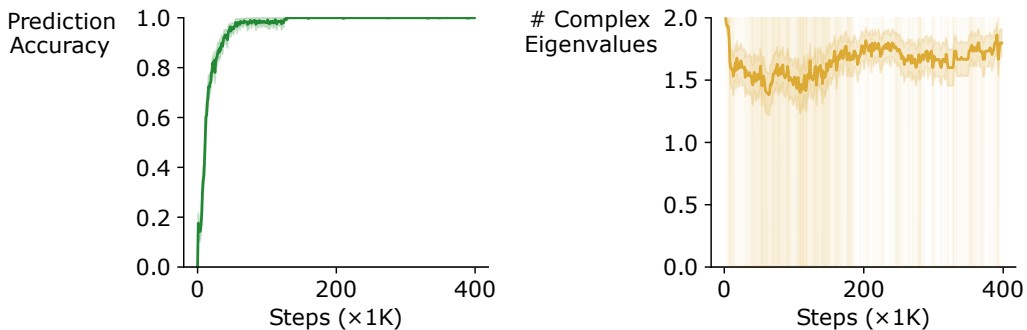

Figure 8: **Left**: The percentage of correct predictions when training an RNN in the Three State MDP. **Right**: Number of complex eigenvalues when training an RNN in the Three States MDP. The solid line is the mean over 30 runs, the shaded region area is the standard error, and the lines are individual runs.

## C   More on the Equivalence of Non-Linear RTUs and Dense RNNs

As discussed in the main body, we likely only have an equivalence between using a full dense weight matrix and a complex-valued diagonal matrix for linear recurrent layers. However, we can obtain a slightly more general equivalence in the restricted setting where the weight matrix for the recurrence is symmetric. This restricted setting is not of general interest, but we include the result here because it could be of interest to a few.

For the case where we have a symmetric weight matrix, we need the activation to commute with orthonormal matrices. Consider again the form

$$\mathbf{h}_t \doteq f(\mathbf{W}_h \mathbf{h}_{t-1} + \mathbf{W}_x \mathbf{x}_t) \tag{14}$$

where $f$ is a potentially nonlinear function that inputs a vector and outputs the same-sized vector. To obtain the equivalence, assume that for any orthonormal matrix $\mathbf{A}$, $f \circ \mathbf{A} = \mathbf{A} \circ f$. We can rewrite the square and symmetric matrix $\mathbf{W}_h$ using an eigenvalue decomposition $\mathbf{W}_h = \mathbf{A} \boldsymbol{\Lambda} \mathbf{A}^\top$, where $\mathbf{A}$ contains the $n$ linearly independent eigenvectors and is an orthonormal matrix and $\boldsymbol{\Lambda}$ is a diagonal matrix with the corresponding eigenvalues. Then, we can re-write (14) as:

$$
\begin{aligned}
\mathbf{h}_t &= f(\mathbf{A} \boldsymbol{\Lambda} \mathbf{A}^\top \mathbf{h}_{t-1} + \mathbf{A}\mathbf{A}^\top \mathbf{W}_x \mathbf{x}_t) \\
&= \mathbf{A} f(\boldsymbol{\Lambda} \mathbf{A}^\top \mathbf{h}_{t-1} + \mathbf{A}^\top \mathbf{W}_x \mathbf{x}_t) \\
\mathbf{A}^\top \mathbf{h}_t &= \boldsymbol{\Lambda} \mathbf{A}^\top \mathbf{h}_{t-1} + \mathbf{A}^\top \mathbf{W}_x \mathbf{x}_t
\end{aligned}
\tag{17}
$$

where $\mathbf{A}$ came outside of $f$ because commutes with orthonormal matrices. By defining $\overline{\mathbf{h}}_t \doteq \mathbf{A}^\top \mathbf{h}_t$ and $\overline{\mathbf{W}}_x \doteq \mathbf{A}^\top \mathbf{W}_x$, we get:

$$\overline{\mathbf{h}}_t = f(\boldsymbol{\Lambda}\overline{\mathbf{h}}_{t-1} + \overline{\mathbf{W}}_x \mathbf{x}_t). \tag{15}$$

Each hidden unit now has one recurrent relation instead of n, because our weight matrix $\boldsymbol{\Lambda}$ is diagonal.

A natural question is if only the identity $f(\mathbf{x}) = \mathbf{x}$—namely linear recurrence—satisfies this property of commuting with orthonormal matrices. We show below that it holds for a slightly more general class of recurrent layers, proving an if-and-only-if relationship. We see the below result as a negative result, highlighting that this equivalence largely only holds for the linear setting and does not generalize to other activations of interest. It provides even further evidence that likely the only setting of interest for the general non-symmetric case is also with a linear recurrence.

Nonetheless, let us obtain the if-and-only-if for completeness. A simple extension that continues to satisfies this property is $f(\mathbf{x}) = \mathbf{x}c(||\mathbf{x}||_2)$ for any $c : \mathbb{R} \to \mathbb{R}$. We can see that for any orthonormal $\mathbf{A}$, we have

$$f(\mathbf{A}\mathbf{x}) = \mathbf{A}\mathbf{x}c(||\mathbf{A}\mathbf{x}||_2) = \mathbf{A}\mathbf{x}c(||\mathbf{x}||_2) = \mathbf{A}f(\mathbf{x}).$$

This means that we can have activations that rescale the input $\mathbf{x}$ depending on the norm of that input.

More generally, the activation can involve matrices and rotations. We can define $\boldsymbol{\theta}(\mathbf{x}) = [\mathbf{x}\ \mathbf{U}(\mathbf{x})]^\top$ where $\mathbf{U}(\mathbf{x}) \in \mathbb{R}^{n \times n-1}$ is a matrix where the columns are orthogonal vectors to each other and to $\mathbf{x}$. Then for any vector-valued $\mathbf{g} : \mathbb{R} \to \mathbb{R}^n$, we have that $f(\mathbf{x}) = \boldsymbol{\theta}(\mathbf{x})^\top g(||\mathbf{x}||_2)$ also satisfies this property:

$$
\begin{aligned}
f(\mathbf{A}\mathbf{x}) &= \theta(\mathbf{A}\mathbf{x})^\top g(||\mathbf{A}\mathbf{x}||_2) = [\mathbf{A}\mathbf{x}; U(\mathbf{A}\mathbf{x})]g(||\mathbf{x}||_2) \\
&= \mathbf{A}[\mathbf{x}; \mathbf{U}(\mathbf{x})]g(||\mathbf{x}||_2) = \mathbf{A}f(\mathbf{x}).
\end{aligned}
$$

The last line follows because $\mathbf{A}\mathbf{U}(\mathbf{x}) = \mathbf{U}(\mathbf{A}\mathbf{x})$. Namely, for any orthogonal vector $\mathbf{u}$ with $\mathbf{u}^\top \mathbf{x} = 0$, we have that $\tilde{\mathbf{u}} \doteq \mathbf{A}\mathbf{u}$ satisfies $\tilde{\mathbf{u}}^\top \mathbf{A}\mathbf{x} = \mathbf{u}^\top \mathbf{A}^\top \mathbf{A}\mathbf{x} = \mathbf{u}^\top \mathbf{x} = 0$ because $\mathbf{A}^\top \mathbf{A} = \mathbf{I}$.

Now we show this formally. Denote $O^n \subset \mathbb{R}^{n \times n} : A \in O^n \iff A^T A = A A^T = I$. Denote $e_i = \begin{bmatrix} 0 \\ \vdots \\ 1 \\ \vdots \\ 0 \end{bmatrix}$, where the $i^{th}$ element is 1.

**Definition C.1.** $\boldsymbol{\theta}(\mathbf{x})$ is a matrix such that:

$$
\begin{aligned}
\boldsymbol{\theta}(\mathbf{x}) &\in O^n \\
\boldsymbol{\theta}(\mathbf{x})\mathbf{x} &= ||\mathbf{x}||_2^2 e_1
\end{aligned}
\tag{18}
$$

**Lemma C.2.** $\boldsymbol{\theta}(\mathbf{x})$ *exits and* $\forall \mathbf{A} \in O^n, \boldsymbol{\theta}(\mathbf{A}\mathbf{x}) = \boldsymbol{\theta}(\mathbf{x})\mathbf{A}^T$

*Proof.* Let $\boldsymbol{\theta}(\mathbf{x})$ be $\in O^n$. Then, by definition of $O^n$, $\boldsymbol{\theta}(\mathbf{x})^T\boldsymbol{\theta}(\mathbf{x}) = \boldsymbol{\theta}(\mathbf{x})\boldsymbol{\theta}(\mathbf{x})^T = I$. Let $\boldsymbol{\theta}(\mathbf{x})[1] = \mathbf{x}$, and $\boldsymbol{\theta}(\mathbf{x})[2:]$ be any orthogonal vectors, also orthogonal to $\mathbf{x}$. Where $\boldsymbol{\theta}(\mathbf{x})[i]$ be the $i^{th}$ row of $\boldsymbol{\theta}(\mathbf{x})$. Then, for $\mathbf{x} \in \mathbb{R}^n$, $\boldsymbol{\theta}(\mathbf{x})\mathbf{x} = \|\mathbf{x}\|_2 e_1$.

$$\boldsymbol{\theta}(\mathbf{Ax})\mathbf{Ax} = \|\mathbf{Ax}\|_2{}^2 e_1 = \|\mathbf{x}\|_2{}^2 e_1 = \boldsymbol{\theta}(\mathbf{x})\mathbf{x}$$
$$\boldsymbol{\theta}(\mathbf{Ax})^T\boldsymbol{\theta}(\mathbf{Ax})\mathbf{Ax} = \boldsymbol{\theta}(\mathbf{Ax})^T\boldsymbol{\theta}(\mathbf{x})\mathbf{x}$$
$$\mathbf{Ax} = \boldsymbol{\theta}(\mathbf{Ax})^T\boldsymbol{\theta}(\mathbf{x})\mathbf{x}$$
$$\mathbf{A}^T\mathbf{Ax} = \mathbf{A}^T\boldsymbol{\theta}(\mathbf{Ax})^T\boldsymbol{\theta}(\mathbf{x})\mathbf{x} \tag{19}$$
$$\mathbf{x} = \mathbf{A}^T\boldsymbol{\theta}(\mathbf{Ax})^T\boldsymbol{\theta}(\mathbf{x})\mathbf{x}$$
$$\mathbf{A}^T\boldsymbol{\theta}(\mathbf{Ax})^T\boldsymbol{\theta}(\mathbf{x}) = I$$
$$\boldsymbol{\theta}(\mathbf{Ax})^T = \boldsymbol{\theta}(\mathbf{x})\mathbf{A}^T$$

$\square$

**Theorem C.3.** *Let* $f : \mathbb{R}^n \to \mathbb{R}^n$ *and* $A \in O^n$. *Then,* $f \circ A = A \circ f \iff \exists g : \mathbb{R} \to \mathbb{R}^n, f(x) = \theta(x)^T g(\|x\|_2)$

*Proof.* Define $g(\alpha) = f(\alpha^2 e_1)$.

$$\begin{aligned}
\mathbf{f}(\mathbf{x}) &= \boldsymbol{\theta}(\mathbf{x})^T\boldsymbol{\theta}(\mathbf{x})\mathbf{f}(\mathbf{x}) \\
&= \boldsymbol{\theta}(\mathbf{x})^T\mathbf{f}(\boldsymbol{\theta}(\mathbf{x})\mathbf{x}) \\
&= \boldsymbol{\theta}(\mathbf{x})^T\mathbf{f}(\|\mathbf{x}\|_\mathbf{2}{}^\mathbf{2}\mathbf{e_1}) \\
&= \boldsymbol{\theta}(\mathbf{x})^T\mathbf{g}(\|\mathbf{x}\|_2)
\end{aligned} \tag{20}$$

$$\begin{aligned}
(\mathbf{f} \circ \mathbf{A})(\mathbf{x}) &= \mathbf{f}(\mathbf{Ax}) \\
&= \boldsymbol{\theta}(\mathbf{Ax})^T\mathbf{g}(\|\mathbf{Ax}\|_2) \\
&= \boldsymbol{\theta}(\mathbf{Ax})^T\mathbf{g}(\|\mathbf{x}\|_2) \\
&= \mathbf{A}\boldsymbol{\theta}(\mathbf{x})^T\mathbf{g}(\|\mathbf{x}\|_2) \\
&= (\mathbf{A} \circ \mathbf{f})(\mathbf{x})
\end{aligned} \tag{21}$$

$\square$

# D   Issues with Two Alternative Parameterizations

In this section we provide a few additional insights on alternative ways to handle complex numbers within an RNN, and why they are not preferable.

## D.1   Stability

We look at the gradient when using each complex representation to understand how different representations affect learning stability. Since each hidden unit has only one recurrent relation in diagonal RNNs, it is sufficient to consider one unit, in isolation. To keep the below intuition simple, we also omit the input of $\mathbf{x}$, and consider

$$h_t = \lambda h_{t-1} = \ldots = \lambda^t h_0 \tag{22}$$

where $h_0$ is the initial hidden state.

**Real Representation** $a + bi$**:** Substituting $\lambda$ in (22) with the real representation, we get:

$$h_t = (a + bi)^t h_0 = h_0 \sum_{k=0}^{t} \binom{t}{k} a^{t-k} b^k i^k$$

Then it follows that the gradient w.r.t the learnable parameters $a$ and $b$ is:

$$\frac{\partial h_t}{\partial a} = h_0 \sum_{k=0}^{t} \binom{t}{k} (t-k) a^{t-k-1} b^k i^k$$

$$\frac{\partial h_t}{\partial b} = h_0 \sum_{k=0}^{t} \binom{t}{k} k a^{t-k} b^{k-1} i^k$$

To prevent the gradient from vanishing/exploding, we need to restrict both $|a|$ and $|b|$ to be $\in (0, 1]$.

**Exponential Representation** $r \exp(i\theta)$**:** Substituting $\lambda$ in (22) with the exponential representation, we get:

$$h_t = r^t \exp(it\theta) h_0$$

and the gradient w.r.t the learnable parameters $r$ and $\theta$ is:

$$\frac{\partial h_t}{\partial r} = t r^{t-1} \exp(it\theta) h_0, \quad \frac{\partial h_t}{\partial \theta} = r^t \exp(it\theta) it h_0$$

To prevent the gradient from vanishing/exploding, we need to restrict $r \in (0, 1]$.

**Cosine Representation** $r(\cos(\theta) + i \sin(\theta))$**:** Substituting $\lambda$ in (22) with the cosine representation, we get:

$$h_t = r^t (\cos(t\theta) + i \sin(t\theta)) h_0$$

and the gradient w.r.t the learnable parameters $r$ and $\theta$ is:

$$\frac{\partial h_t}{\partial r} = t r^{t-1} (\cos(t\theta) + i \sin(t\theta)) h_0$$

$$\frac{\partial h_t}{\partial \theta} = r^t (it \cos(t\theta) - t \sin t\theta) h_0.$$

To prevent the gradient from vanishing/exploding, we need to restrict $r \in (0, 1]$. It is simpler to maintain stability with the exponential and cosine representations, since we only need to constrain $r \in (0, 1]$. , whereas the real representation requires us to restrict both the complex number's magnitude and phase.

### D.2 Biased gradient when only using the real part of the hidden state

We can attempt to get the benefits of having a real-valued hidden state by simply converting the complex-valued state to a real-valued one within the LRU. However, taking only the real part results in a biased gradient, as we show in this section.

Consider again the one recurrent unit example, but now also consider the output obtained by taking only the real part of the recurrent state: $y_t = wRe\{h_t\}$, where $w$ is a learnable parameter. The gradient w.r.t $r$ and $\theta$ is:

$$\frac{\partial y_t}{\partial r} = w \left( \cos(\theta) h_{t-1} + r \cos(\theta) \frac{\partial h_{t-1}}{\partial r} \right)$$

$$\frac{\partial y_t}{\partial \theta} = w \left( -r \sin(\theta) h_{t-1} + r \cos(\theta) \frac{\partial h_{t-1}}{\partial \theta} \right)$$

$$(23)$$

Multiplying by a complex number $z$ is equivalent to a rotation by the matrix $\begin{bmatrix} Re\{z\} & -Img\{z\} \\ Img\{z\} & Re\{z\} \end{bmatrix}$ and a scale by $\sqrt{Re\{z\}^2 + Img\{z\}^2}$. We re-write the recurrent unit using this property as:

$$h_t^{c_1} = r \cos(\theta) h_{t-1}^{c_1} - r \sin(\theta) h_{t-1}^{c_2}$$

$$h_t^{c_2} = r \cos(\theta) h_{t-1}^{c_2} + r \sin(\theta) h_{t-1}^{c_1}$$

$$y_t = w(h_t^{c_1} + h_t^{c_2})$$

$$(24)$$

Notice that we don't need to take the real part of the recurrent state in this formulation. We now write the gradient using this new formulation:

$$
\begin{aligned}
\frac{\partial y_t}{\partial r} &= w(r\cos(\theta)(h^{c_1}_{t-1} + h^{c_2}_{t-1}) + r\cos(\theta)(\frac{\partial h^{c_1}_{t-1}}{\partial r} + \frac{\partial h^{c_2}_{t-1}}{\partial r}) \\
&\quad + r\sin(\theta)(h^{c_1}_{t-1} - h^{c_2}_{t-1}) + r\sin(\theta)(\frac{\partial h^{c_1}_{t-1}}{\partial r} - \frac{\partial h^{c_2}_{t-1}}{\partial r})) \\
\frac{\partial y_t}{\partial \theta} &= w(-r\sin(\theta)(h^{c_1}_{t-1} + h^{c_2}_{t-1}) + r\cos(\theta)(\frac{\partial h^{c_1}_{t-1}}{\partial \theta} + \frac{\partial h^{c_2}_{t-1}}{\partial \theta}) \\
&\quad + r\cos(\theta)(h^{c_1}_{t-1} - h^{c_2}_{t-1}) + r\sin(\theta)(\frac{\partial h^{c_1}_{t-1}}{\partial \theta} - \frac{\partial h^{c_2}_{t-1}}{\partial \theta}))
\end{aligned}
\tag{25}
$$

Comparing Eq. 25 and Eq. 23, we can see that using only the real part of the recurrent state leads to a loss of information in the gradient.

## E    Recurrent Trace Units

This appendix details the parametrization used for RTUs, the derivation of the RTRL update rules, and the extension of RTUs to multi-layers.

### E.1    Empirical Analysis for different $r$ and $\theta$ parameterizations:

Since $r$ represents the magnitude of a complex number, then $r \in \mathbb{R}^+$, it is preferred to have $r \in (0, 1]$ to avoid vanishing/exploding gradients as discussed in the previous section. Let $w_r$ be a learnable parameter which could directly represent $r$ or represent a function of $r$, we can enforce the constraints on $r$ in several ways:

1. Direct learning: Learn $r$ directly, and clip it after each parameter's update to be in $(0, 1]$.
2. Enforcing $r \in \mathbb{R}^+$: Learn $\nu$ such that $r \doteq \exp(-\nu)$. This parameterization enforces $r$ to be positive. However, additional clipping is needed to enforce $r \in (0, 1]$.
3. Enforcing stability on $r$: We can enforce $r$ to be $\in (0, 1]$ by using a positive non-linear function. For example, learn $\nu^{\log}$ such that $r \doteq \exp(-\exp(\nu^{\log}))$, this parameterization ensures that $r \in (0, 1]$ and is suggested by Orvieto et al. [34]. Another example is learning $r = \sigma(\nu)$, which also ensures $r \in (0, 1]$.

Finally, we can also enforce stability on $\theta$ by learning $\theta^{\log}$ such that $\theta \doteq \exp(\theta^{\log})$ to ensure that $\theta$ is always positive.

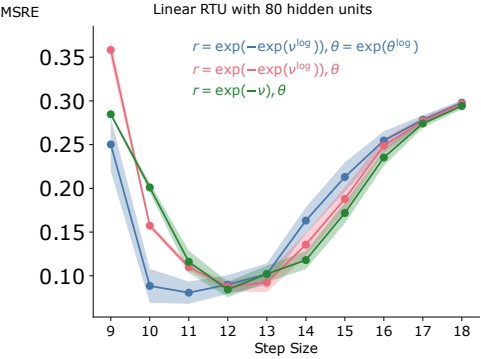

Figure 9: Learning rate sensitivity for different parameterizations of $r$ and $\theta$ for RTUs with $80$ hidden units.

We empirically compare the different parameterizations of $r$ and $\theta$. In our experiments, learning $r$ directly resulted in unstable training where the MSRE diverges. We plot the learning rate sensitivity

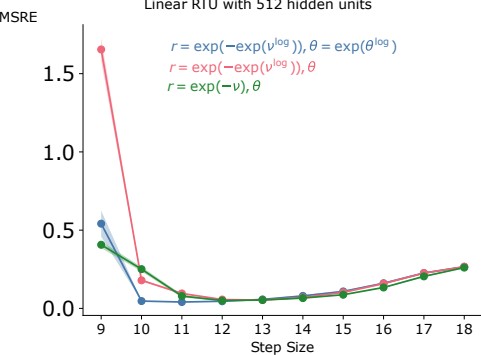

Figure 10: Learning rate sensitivity for different parameterizations of $r$ and $\theta$ for RTUs with 512 hidden units.

for the other parameterization for two different sizes of RTUs in Figures 9 and 10. While all the parametrizations produce similar performance, we notice that learning $\theta^{\log}$ and $\nu^{\log}$ has better learning rate sensitivity.

### E.2 Real-Time Recurrent Learning for a Single Layer Linear RTUs

We now outline the details for the RTRL update rules for RTUs. The set of learnable parameters for RTUs is $\boldsymbol{\psi} \doteq \{\boldsymbol{\nu}^{\log}, \boldsymbol{\theta}^{\log}, \mathbf{W}_x^{c_1}, \mathbf{W}_x^{c_2}\}$. At each time step $t$, the learner receives a loss $\mathcal{L}_t(\hat{y}_t, y_t; \boldsymbol{\psi})$ where $y_t$ is the network output at time $t$, then the gradient of the loss w.r.t the parameters is:

$$\frac{\partial \mathcal{L}_t}{\partial \boldsymbol{\psi}} = \frac{\partial \mathcal{L}_t}{\partial \mathbf{h}_t} \frac{\partial \mathbf{h}_t}{\partial \mathbf{h}_t^{c_1}} \frac{\partial \mathbf{h}_t^{c_1}}{\partial \boldsymbol{\psi}} + \frac{\partial \mathcal{L}_t}{\partial \mathbf{h}_t} \frac{\partial \mathbf{h}_t}{\partial \mathbf{h}_t^{c_2}} \frac{\partial \mathbf{h}_t^{c_2}}{\partial \boldsymbol{\psi}}, \tag{26}$$

where $\frac{\partial \mathbf{h}_t^{c_1}}{\partial \boldsymbol{\psi}} = \left\{ \frac{\partial \mathbf{h}_t^{c_1}}{\partial \boldsymbol{\nu}^{\log}}, \frac{\partial \mathbf{h}_t^{c_1}}{\partial \boldsymbol{\theta}^{\log}}, \frac{\partial \mathbf{h}_t^{c_1}}{\partial \boldsymbol{W}_x^{c_1}}, \frac{\partial \mathbf{h}_t^{c_1}}{\partial \boldsymbol{W}_x^{c_1}} \right\}$ and $\frac{\partial \mathbf{h}_t^{c_2}}{\partial \boldsymbol{\psi}} = \left\{ \frac{\partial \mathbf{h}_t^{c_2}}{\partial \boldsymbol{\nu}^{\log}}, \frac{\partial \mathbf{h}_t^{c_2}}{\partial \boldsymbol{\theta}^{\log}}, \frac{\partial \mathbf{h}_t^{c_2}}{\partial \boldsymbol{W}_x^{c_2}}, \frac{\partial \mathbf{h}_t^{c_2}}{\partial \boldsymbol{W}_x^{c_2}} \right\}$.

We can derive the following gradients:

$$\frac{\partial \mathbf{h}_t^{c_1}}{\partial \boldsymbol{\nu}^{\log}} = \frac{\partial \mathbf{g}(\boldsymbol{\nu}, \boldsymbol{\theta})}{\partial \boldsymbol{\nu}^{\log}} \odot \mathbf{h}_{t-1}^{c_1} + \mathbf{g}(\boldsymbol{\nu}, \boldsymbol{\theta}) \frac{\partial \mathbf{h}_{t-1}^{c_1}}{\partial \boldsymbol{\nu}^{\log}} - \frac{\partial \boldsymbol{\phi}(\boldsymbol{\nu}, \boldsymbol{\theta})}{\partial \boldsymbol{\nu}^{\log}} \odot \mathbf{h}_{t-1}^{c_2} - \boldsymbol{\phi}(\boldsymbol{\nu}, \boldsymbol{\theta}) \frac{\partial \mathbf{h}_{t-1}^{c_2}}{\partial \boldsymbol{\nu}^{\log}} + \frac{\partial \boldsymbol{\gamma}}{\partial \boldsymbol{\nu}_{\log}} \odot \mathbf{W}_x^{c_1} \mathbf{x}_t$$

$$\frac{\partial \mathbf{h}_t^{c_2}}{\partial \boldsymbol{\nu}^{\log}} = \frac{\partial \mathbf{g}(\boldsymbol{\nu}, \boldsymbol{\theta})}{\partial \boldsymbol{\nu}^{\log}} \odot \mathbf{h}_{t-1}^{c_2} + \mathbf{g}(\boldsymbol{\nu}, \boldsymbol{\theta}) \frac{\partial \mathbf{h}_{t-1}^{c_1}}{\partial \boldsymbol{\nu}^{\log}} + \frac{\partial \boldsymbol{\phi}(\boldsymbol{\nu}, \boldsymbol{\theta})}{\partial \boldsymbol{\nu}^{\log}} \odot \mathbf{h}_{t-1}^{c_1} + \boldsymbol{\phi}(\boldsymbol{\nu}, \boldsymbol{\theta}) \frac{\partial \mathbf{h}_{t-1}^{c_1}}{\partial \boldsymbol{\nu}^{\log}} + \frac{\partial \boldsymbol{\gamma}}{\partial \boldsymbol{\nu}_{\log}} \odot \mathbf{W}_x^{c_2} \mathbf{x}_t \tag{27}$$

$$\frac{\partial \mathbf{h}_t^{c_1}}{\partial \boldsymbol{\theta}^{\log}} = \frac{\partial \mathbf{g}(\boldsymbol{\nu}, \boldsymbol{\theta})}{\partial \boldsymbol{\theta}^{\log}} \odot \mathbf{h}_{t-1}^{c_1} + \mathbf{g}(\boldsymbol{\nu}, \boldsymbol{\theta}) \frac{\partial \mathbf{h}_{t-1}^{c_1}}{\partial \boldsymbol{\theta}^{\log}} - \frac{\partial \boldsymbol{\phi}(\boldsymbol{\nu}, \boldsymbol{\theta})}{\partial \boldsymbol{\theta}^{\log}} \odot \mathbf{h}_{t-1}^{c_2} - \boldsymbol{\phi}(\boldsymbol{\nu}, \boldsymbol{\theta}) \frac{\partial \mathbf{h}_{t-1}^{c_2}}{\partial \boldsymbol{\theta}^{\log}}$$

$$\frac{\partial \mathbf{h}_t^{c_2}}{\partial \boldsymbol{\theta}^{\log}} = \frac{\partial \mathbf{g}(\boldsymbol{\nu}, \boldsymbol{\theta})}{\partial \boldsymbol{\theta}^{\log}} \odot \mathbf{h}_{t-1}^{c_2} + \mathbf{g}(\boldsymbol{\nu}, \boldsymbol{\theta}) \frac{\partial \mathbf{h}_{t-1}^{c_2}}{\partial \boldsymbol{\theta}^{\log}} + \frac{\partial \boldsymbol{\phi}(\boldsymbol{\nu}, \boldsymbol{\theta})}{\partial \boldsymbol{\theta}^{\log}} \odot \mathbf{h}_{t-1}^{c_1} + \boldsymbol{\phi}(\boldsymbol{\nu}, \boldsymbol{\theta}) \frac{\partial \mathbf{h}_{t-1}^{c_1}}{\partial \boldsymbol{\theta}^{\log}} \tag{28}$$

where

$$\frac{\partial \mathbf{g}(\boldsymbol{\nu}, \boldsymbol{\theta})}{\partial \boldsymbol{\nu}^{\log}} = -\mathbf{g}(\boldsymbol{\nu}, \boldsymbol{\theta}) \exp(\boldsymbol{\nu}^{\log})$$

$$\frac{\partial \mathbf{g}(\boldsymbol{\nu}, \boldsymbol{\theta})}{\partial \boldsymbol{\theta}^{\log}} = -\boldsymbol{\phi}(\boldsymbol{\nu}, \boldsymbol{\theta}) \exp(\boldsymbol{\theta}^{\log})$$

$$\frac{\partial \boldsymbol{\phi}(\boldsymbol{\nu}, \boldsymbol{\theta})}{\partial \boldsymbol{\nu}^{\log}} = -\boldsymbol{\phi}(\boldsymbol{\nu}, \boldsymbol{\theta}) \exp(\boldsymbol{\nu}^{\log}) \tag{29}$$

$$\frac{\partial \boldsymbol{\phi}(\boldsymbol{\nu}, \boldsymbol{\theta})}{\partial \boldsymbol{\theta}^{\log}} = \mathbf{g}(\boldsymbol{\nu}, \boldsymbol{\theta}) \exp(\boldsymbol{\theta}^{\log})$$

To efficiently compute the gradient w.r.t $\mathbf{W}_x^{c_1}$ and $\mathbf{W}_x^{c_2}$, we look at the influence of each when considering a single element from each recurrent state, $\mathbf{h}_t^{c_1}$ and $\mathbf{h}_t^{c_2}$:

$$h_{t,i}^{c_1} = g(\nu_i, \theta_i) h_{t-1,i}^{c_1} - \phi(\nu_i, \theta_i) h_{t-1,i}^{c_2} + \gamma_i \sum_{j=0}^{d} w_{x,(i,j)}^{c_1} x_{t,j}$$

$$h_{t,i}^{c_2} = g(\nu_i, \theta_i) h_{t-1,i}^{c_2} + \phi(\nu_i, \theta_i) h_{t-1,i}^{c_1} + \gamma_i \sum_{j=0}^{d} w_{x,(i,j)}^{c_2} x_{t,j}.$$

(30)

We then get:

$$\frac{\partial h_{t,i}^{c_1}}{\partial W_{x,(i,j)}^{c_1}} = g(\nu_i, \theta_i) \frac{\partial h_{t-1,i}^{c_1}}{\partial W_{x,(i,j)}^{c_1}} - \phi(\nu_i, \theta_i) \frac{\partial h_{t-1,i}^{c_2}}{\partial W_{x,(i,j)}^{c_1}} + \gamma_i x_{t,j}$$

$$\frac{\partial h_{t,i}^{c_2}}{\partial W_{x,(i,j)}^{c_1}} = g(\nu_i, \theta_i) \frac{\partial h_{t-1,i}^{c_2}}{\partial W_{x,(i,j)}^{c_1}} + \phi(\nu_i, \theta_i) \frac{\partial h_{t-1,i}^{c_1}}{\partial W_{x,(i,j)}^{c_1}}$$

$$\frac{\partial h_{t,i}^{c_1}}{\partial W_{x,(i,j)}^{c_2}} = g(\nu_i, \theta_i) \frac{\partial h_{t-1,i}^{c_1}}{\partial W_{x,(i,j)}^{c_2}} - \phi(\nu_i, \theta_i) \frac{\partial h_{t-1,i}^{c_2}}{\partial W_{x,(i,j)}^{c_2}}$$

$$\frac{\partial h_{t,i}^{c_2}}{\partial W_{x,(i,j)}^{c_2}} = g(\nu_i, \theta_i) \frac{\partial h_{t-1,i}^{c_2}}{\partial W_{x,(i,j)}^{c_2}} + \phi(\nu_i, \theta_i) \frac{\partial h_{t-1,i}^{c_1}}{\partial W_{x,(i,j)}^{c_2}} + \gamma_i x_{t,j}.$$

(31)

We see that each $h_{t,i}^{c_1}$ gets affected by weights from only one row of $\mathbf{W}_x^{c_1}$, thus, $\frac{\partial \mathbf{h}_t^{c_1}}{\mathbf{W}_x^{c_1}}$ can be written as a matrix of the same dimension as $\mathbf{W}_x^{c_1}$. The same is true for $\frac{\partial \mathbf{h}_t^{c_2}}{\mathbf{W}_x^{c_2}}, \frac{\partial \mathbf{h}_t^{c_1}}{\mathbf{W}_x^{c_2}}$, and $\frac{\partial \mathbf{h}_t^{c_2}}{\mathbf{W}_x^{c_1}}$.

### E.3 Single Layer Non-Linear RTUs Formulation:

We extend the linear RTUs to non-linear RTUs by adding a non-linear activation function $\mathbf{f}$ to the recurrent states. We can write the non-linear RTUs as follows:

$$\mathbf{h}_t^{c_1} = \mathbf{f}(\mathbf{g}(\boldsymbol{\nu}, \boldsymbol{\theta}) \odot \mathbf{h}_{t-1}^{c_1} - \boldsymbol{\phi}(\boldsymbol{\nu}, \boldsymbol{\theta}) \odot \mathbf{h}_{t-1}^{c_2} + \boldsymbol{\gamma} \odot \mathbf{W}_x^{c_1} \mathbf{x}_t)$$
$$\mathbf{h}_t^{c_2} = \mathbf{f}(\mathbf{g}(\boldsymbol{\nu}, \boldsymbol{\theta}) \odot \mathbf{h}_{t-1}^{c_2} + \boldsymbol{\phi}(\boldsymbol{\nu}, \boldsymbol{\theta}) \odot \mathbf{h}_{t-1}^{c_1} + \boldsymbol{\gamma} \odot \mathbf{W}_x^{c_2} \mathbf{x}_t)$$
$$\mathbf{h}_t = [\mathbf{h}_t^{c_1}; \mathbf{h}_t^{c_2}]$$

(32)

where $\mathbf{f} : \mathbb{R}^n \to \mathbb{R}^n$. Following the same procedure as in the linear case, we can derive RTRL update rules for the non-linear RTUs.

$$\frac{\partial \mathbf{h}_t^{c_1}}{\partial \boldsymbol{\nu}^{\log}} = \mathbf{f}'(\cdot)(\frac{\partial \mathbf{g}(\boldsymbol{\nu}, \boldsymbol{\theta})}{\partial \boldsymbol{\nu}^{\log}} \odot \mathbf{h}_{t-1}^{c_1} + \mathbf{g}(\boldsymbol{\nu}, \boldsymbol{\theta}) \frac{\partial \mathbf{h}_{t-1}^{c_1}}{\partial \boldsymbol{\nu}^{\log}} - \frac{\partial \boldsymbol{\phi}(\boldsymbol{\nu}, \boldsymbol{\theta})}{\partial \boldsymbol{\nu}^{\log}} \odot \mathbf{h}_{t-1}^{c_2} - \boldsymbol{\phi}(\boldsymbol{\nu}, \boldsymbol{\theta}) \frac{\partial \mathbf{h}_{t-1}^{c_2}}{\partial \boldsymbol{\nu}^{\log}})$$
$$\frac{\partial \mathbf{h}_t^{c_2}}{\partial \boldsymbol{\nu}^{\log}} = \mathbf{f}'(\cdot)(\frac{\partial \mathbf{g}(\boldsymbol{\nu}, \boldsymbol{\theta})}{\partial \boldsymbol{\nu}^{\log}} \odot \mathbf{h}_{t-1}^{c_2} + \mathbf{g}(\boldsymbol{\nu}, \boldsymbol{\theta}) \frac{\partial \mathbf{h}_{t-1}^{c_2}}{\partial \boldsymbol{\nu}^{\log}} + \frac{\partial \boldsymbol{\phi}(\boldsymbol{\nu}, \boldsymbol{\theta})}{\partial \boldsymbol{\nu}^{\log}} \odot \mathbf{h}_{t-1}^{c_1} + \boldsymbol{\phi}(\boldsymbol{\nu}, \boldsymbol{\theta}) \frac{\partial \mathbf{h}_{t-1}^{c_1}}{\partial \boldsymbol{\nu}^{\log}})$$

(33)

$$\frac{\partial \mathbf{h}_t^{c_1}}{\partial \boldsymbol{\theta}^{\log}} = \mathbf{f}'(\cdot)(\frac{\partial \mathbf{g}(\boldsymbol{\nu}, \boldsymbol{\theta})}{\partial \boldsymbol{\theta}^{\log}} \odot \mathbf{h}_{t-1}^{c_1} + \mathbf{g}(\boldsymbol{\nu}, \boldsymbol{\theta}) \frac{\partial \mathbf{h}_{t-1}^{c_1}}{\partial \boldsymbol{\theta}^{\log}} - \frac{\partial \boldsymbol{\phi}(\boldsymbol{\nu}, \boldsymbol{\theta})}{\partial \boldsymbol{\theta}^{\log}} \odot \mathbf{h}_{t-1}^{c_2} - \boldsymbol{\phi}(\boldsymbol{\nu}, \boldsymbol{\theta}) \frac{\partial \mathbf{h}_{t-1}^{c_2}}{\partial \boldsymbol{\theta}^{\log}})$$
$$\frac{\partial \mathbf{h}_t^{c_2}}{\partial \boldsymbol{\theta}^{\log}} = \mathbf{f}'(\cdot)(\frac{\partial \mathbf{g}(\boldsymbol{\nu}, \boldsymbol{\theta})}{\partial \boldsymbol{\theta}^{\log}} \odot \mathbf{h}_{t-1}^{c_2} + \mathbf{g}(\boldsymbol{\nu}, \boldsymbol{\theta}) \frac{\partial \mathbf{h}_{t-1}^{c_2}}{\partial \boldsymbol{\theta}^{\log}} + \frac{\partial \boldsymbol{\phi}(\boldsymbol{\nu}, \boldsymbol{\theta})}{\partial \boldsymbol{\theta}^{\log}} \odot \mathbf{h}_{t-1}^{c_1} + \boldsymbol{\phi}(\boldsymbol{\nu}, \boldsymbol{\theta}) \frac{\partial \mathbf{h}_{t-1}^{c_1}}{\partial \boldsymbol{\theta}^{\log}})$$

(34)

$$h_{t,i}^{c_1} = f(g(\nu_i, \theta_i) h_{t-1,i}^{c_1} - \phi(\nu_i, \theta_i) h_{t-1,i}^{c_2} + \gamma_i \sum_{j=0}^{d} w_{x,(i,j)}^{c_1} x_{t,j})$$

$$h_{t,i}^{c_2} = f(g(\nu_i, \theta_i) h_{t-1,i}^{c_2} + \phi(\nu_i, \theta_i) h_{t-1,i}^{c_1} + \gamma_i \sum_{j=0}^{d} w_{x,(i,j)}^{c_2} x_{t,j})$$

(35)

We then get:

$$
\begin{aligned}
\frac{\partial h_{t,i}^{c_1}}{\partial W_{x,(i,j)}^{c_1}} &= f'(\cdot)(g(\nu_i,\theta_i)\frac{\partial h_{t-1,i}^{c_1}}{\partial W_{x,(i,j)}^{c_1}} - \phi(\nu_i,\theta_i)\frac{\partial h_{t-1,i}^{c_2}}{\partial W_{x,(i,j)}^{c_1}} + \gamma_i x_{t,j}) \\
\frac{\partial h_{t,i}^{c_2}}{\partial W_{x,(i,j)}^{c_1}} &= f'(\cdot)(g(\nu_i,\theta_i)\frac{\partial h_{t-1,i}^{c_2}}{\partial W_{x,(i,j)}^{c_1}} + \phi(\nu_i,\theta_i)\frac{\partial h_{t-1,i}^{c_1}}{\partial W_{x,(i,j)}^{c_1}}) \\
\frac{\partial h_{t,i}^{c_1}}{\partial W_{x,(i,j)}^{c_2}} &= f'(\cdot)(g(\nu_i,\theta_i)\frac{\partial h_{t-1,i}^{c_1}}{\partial W_{x,(i,j)}^{c_2}} - \phi(\nu_i,\theta_i)\frac{\partial h_{t-1,i}^{c_2}}{\partial W_{x,(i,j)}^{c_2}}) \\
\frac{\partial h_{t,i}^{c_2}}{\partial W_{x,(i,j)}^{c_2}} &= f'(\cdot)(g(\nu_i,\theta_i)\frac{\partial h_{t-1,i}^{c_2}}{\partial W_{x,(i,j)}^{c_2}} + \phi(\nu_i,\theta_i)\frac{\partial h_{t-1,i}^{c_1}}{\partial W_{x,(i,j)}^{c_2}} + \gamma_i x_{t,j})
\end{aligned}
\tag{36}
$$

### E.4 Complexity Analysis of RTUs

We now move to calculate the computation and memory complexity of RTUs when learning using the RTRL rules introduced in the previous section.

For an input $\mathbf{x_t} \in \mathbb{R}^d$ and hidden states $\mathbf{h}_t = [\mathbf{f}(\mathbf{h}_t^{c_1}); \mathbf{f}(\mathbf{h}_t^{c_2})] \in \mathbb{R}^{2n}$, we have $\mathbf{g}(\boldsymbol{\nu},\boldsymbol{\theta}), \boldsymbol{\phi}(\boldsymbol{\nu},\boldsymbol{\theta}), \boldsymbol{\gamma} \in \mathbb{R}^n$ and $\mathbf{W}_x^{c_1}, \mathbf{W}_x^{c_2} \in \mathbb{R}^{d \times n}$.

An agent using the RTU with RTRL needs to store the gradient information, $\frac{\partial \mathbf{h}_{t-1}^{c_1}}{\partial \boldsymbol{\psi}}$ and $\frac{\partial \mathbf{h}_{t-1}^{c_2}}{\partial \boldsymbol{\psi}}$, from one step to the next. We denote the set of saved gradient information as:

$$
\begin{aligned}
\nabla_{\boldsymbol{\nu}^{t-1}} &\doteq \left\{ \frac{\partial \mathbf{h}_{t-1}^{c_1}}{\partial \boldsymbol{\nu}^{\log}}, \frac{\partial \mathbf{h}_{t-1}^{c_2}}{\partial \boldsymbol{\nu}^{\log}} \right\} \\
\nabla_{\boldsymbol{\theta}^{t-1}} &\doteq \left\{ \frac{\partial \mathbf{h}_{t-1}^{c_1}}{\partial \boldsymbol{\theta}^{\log}}, \frac{\partial \mathbf{h}_{t-1}^{c_2}}{\partial \boldsymbol{\theta}^{\log}} \right\} \\
\nabla_{\boldsymbol{W}_x^{t-1}} &\doteq \left\{ \frac{\partial \mathbf{h}_{t-1}^{c_1}}{\partial \mathbf{W}_x^{c_1}}, \frac{\partial \mathbf{h}_{t-1}^{c_2}}{\partial \mathbf{W}_x^{c_1}}, \frac{\partial \mathbf{h}_{t-1}^{c_1}}{\partial \mathbf{W}_x^{c_2}}, \frac{\partial \mathbf{h}_{t-1}^{c_2}}{\partial \mathbf{W}_x^{c_2}} \right\}.
\end{aligned}
\tag{37}
$$

The saved gradient information has the following dimensions:

$$
\begin{aligned}
\nabla_{\boldsymbol{\nu}^{t-1}} &\in \mathbb{R}^{2n} \\
\nabla_{\boldsymbol{\theta}^{t-1}} &\in \mathbb{R}^{2n} \\
\nabla_{\boldsymbol{W}_x^{t-1}} &\in \mathbb{R}^{4(d \times n)}.
\end{aligned}
\tag{38}
$$

Then, it follows that memory complexity for RTU with RTRL is $\mathcal{O}(n+nd)$. i.e., linear in the number of parameters.

For the computational complexity, a forward pass according to 2 has a computational complexity of $\mathcal{O}(n+nd)$. Additionally, after doing the forward pass, the learner needs to update the saved gradient information according to equations 27 through 31 which has a computational complexity of $\mathcal{O}(n+nd)$. To summarize, using Real-Time Recurrent Learning with RTUs has linear computational and memory complexities.

### E.5 Multi-Layers Recurrent Trace Units

We now extend RTUs to a multilayer setting. We show that in the multilayer case, we lose the computational advantages. However, prior work suggested that treating each recurrent layer independently is a sensible choice and allows us to gain a computational advantage[18]. Consider an RTU with $n$ layers, we refer to the hidden dimension of a layer $i$ where $0 < i \le n$ as $d_i$. The network has the following set of parameters:

$$
\begin{aligned}
\boldsymbol{\psi} &\doteq \{\boldsymbol{\psi}_1, \boldsymbol{\psi}_2, \boldsymbol{\psi}_3, \ldots, \boldsymbol{\psi}_n\}, \\
\boldsymbol{\psi}_i &\doteq \{\boldsymbol{\nu}^{\log,i}, \boldsymbol{\theta}^{\log,i}, \mathbf{W}_x^{c_1,i}, \mathbf{W}_x^{c_2,i}\}
\end{aligned}
\tag{39}
$$

To update the network parameters, we need to calculate the gradient of the loss w.r.t the parameters from all the layers:

$$\frac{\partial \mathcal{L}_t}{\partial \boldsymbol{\psi}} = \frac{\partial \mathcal{L}_t}{\partial \mathbf{h}_t^n} \frac{\partial \mathbf{h}_t^n}{\partial \boldsymbol{\psi}}$$

$$\frac{\partial \mathbf{h}_t^n}{\partial \boldsymbol{\psi}} = \left\{ \frac{\partial \mathbf{h}_t^n}{\partial \boldsymbol{\psi}_1}, \frac{\partial \mathbf{h}_t^n}{\partial \boldsymbol{\psi}_2}, \frac{\partial \mathbf{h}_t^n}{\partial \boldsymbol{\psi}_3}, \ldots, \frac{\partial \mathbf{h}_t^n}{\partial \boldsymbol{\psi}_n} \right\} \tag{40}$$

where

$$\mathbf{h}_t^n = \begin{bmatrix} \mathbf{f}(\mathbf{h}_t^{c1,n}) \\ \mathbf{f}(\mathbf{h}_t^{c2,n}) \end{bmatrix} \tag{41}$$

Take as an example the gradient of $\mathbf{h}_t^{c1,n}$ w.r.t $\boldsymbol{\nu}^{\log,n}, \boldsymbol{\nu}^{\log,n-1}, \ldots, \boldsymbol{\nu}^{\log,1}$. Unrolling the last two layers of the network, we get:

$$\mathbf{h}_t^{c1,n} = \boldsymbol{g}(\boldsymbol{\nu}^n, \boldsymbol{\theta}^n) \odot \mathbf{h}_{t-1}^{c1,n} - \boldsymbol{\phi}(\boldsymbol{\nu}^n, \boldsymbol{\theta}^n) \odot \mathbf{h}_{t-1}^{c2,n} + \boldsymbol{\gamma}^n \odot \mathbf{W}_x^{c1,n} \mathbf{h}_t^{n-1}$$

$$= \boldsymbol{g}(\boldsymbol{\nu}^n, \boldsymbol{\theta}^n) \odot \mathbf{h}_{t-1}^{c1,n} - \boldsymbol{\phi}(\boldsymbol{\nu}^n, \boldsymbol{\theta}^n) \odot \mathbf{h}_{t-1}^{c2,n} + \boldsymbol{\gamma}^n \odot \mathbf{W}_x^{c1,n} \begin{bmatrix} \mathbf{f}(\mathbf{h}_t^{c1,n-1}) \\ \mathbf{f}(\mathbf{h}_t^{c2,n-1}) \end{bmatrix}$$

$$= \boldsymbol{g}(\boldsymbol{\nu}^n, \boldsymbol{\theta}^n) \odot \mathbf{h}_{t-1}^{c1,n} - \boldsymbol{\phi}(\boldsymbol{\nu}^n, \boldsymbol{\theta}^n) \odot \mathbf{h}_{t-1}^{c2,n} + \boldsymbol{\gamma}^n \odot \mathbf{W}_x^{c1,n}$$

$$\begin{bmatrix} \mathbf{f}(\boldsymbol{g}(\boldsymbol{\nu}^{n-1}, \boldsymbol{\theta}^{n-1}) \odot \mathbf{h}_{t-1}^{c1,n-1} - \boldsymbol{\phi}(\boldsymbol{\nu}^{n-1}, \boldsymbol{\theta}^{n-1}) \odot \mathbf{h}_{t-1}^{c2,n-1} + \boldsymbol{\gamma}^{n-1} \odot \mathbf{W}_x^{c1,n-1} \mathbf{h}_t^{n-2}) \\ \mathbf{f}(\boldsymbol{g}(\boldsymbol{\nu}^{n-1}, \boldsymbol{\theta}^{n-1}) \odot \mathbf{h}_{t-1}^{c2,n-1} + \boldsymbol{\phi}(\boldsymbol{\nu}^{n-1}, \boldsymbol{\theta}^{n-1}) \odot \mathbf{h}_{t-1}^{c1,n-1} + \boldsymbol{\gamma}^{n-1} \odot \mathbf{W}_x^{c2,n-1} \mathbf{h}_t^{n-2}) \end{bmatrix} \tag{42}$$

The gradient of $\mathbf{h}_t^{c1,n}$ w.r.t $\boldsymbol{\nu}^{\log,n}$ can be calculated in linear complexity as indicated in the previous section.

Calculating the gradient w.r.t the parameters from the earlier layer:

$$\frac{\partial \mathbf{h}_t^{c1,n}}{\partial \boldsymbol{\nu}^{\log,n-1}} = \mathbf{g}(\boldsymbol{\nu}^n, \boldsymbol{\theta}^n) \frac{\partial \mathbf{h}_{t-1}^{c1,n}}{\partial \boldsymbol{\nu}^{\log,n-1}} - \boldsymbol{\phi}(\boldsymbol{\nu}^n, \boldsymbol{\theta}^n) \frac{\partial \mathbf{h}_{t-1}^{c2,n}}{\partial \boldsymbol{\nu}^{\log,n-1}} + \boldsymbol{\gamma}^n \odot \mathbf{W}_x^{c1,n} \frac{\partial \mathbf{h}_t^{n-1}}{\partial \boldsymbol{\nu}^{\log,n-1}}$$

$$\frac{\partial \mathbf{h}_t^{n-1}}{\partial \boldsymbol{\nu}^{\log,n-1}} \in \mathbf{R}^{2d_{n-1}} \text{ Can be calulated with linear complexity.}$$

$$\frac{\partial \mathbf{h}_{t-1}^{c1,n}}{\partial \boldsymbol{\nu}^{\log,n-1}} \in \mathbf{R}^{d_n \times d_{n-1}} \text{ Saved from previous timestep.}$$

$$\frac{\partial \mathbf{h}_{t-1}^{c2,n}}{\partial \boldsymbol{\nu}^{\log,n-1}} \in \mathbf{R}^{d_n \times d_{n-1}} \text{ Saved from previous timestep.} \tag{43}$$

$$\frac{\partial \mathbf{h}_t^{c1,n}}{\partial \boldsymbol{\nu}^{\log,n-2}} = \mathbf{g}(\boldsymbol{\nu}^n, \boldsymbol{\theta}^n) \frac{\partial \mathbf{h}_{t-1}^{c1,n}}{\partial \boldsymbol{\nu}^{\log,n-2}} - \boldsymbol{\phi}(\boldsymbol{\nu}^n, \boldsymbol{\theta}^n) \frac{\partial \mathbf{h}_{t-1}^{c2,n}}{\partial \boldsymbol{\nu}^{\log,n-2}} + \boldsymbol{\gamma}^n \odot \mathbf{W}_x^{c1,n} \frac{\partial \mathbf{h}_t^{n-1}}{\partial \boldsymbol{\nu}^{\log,n-2}}$$

$$= \mathbf{g}(\boldsymbol{\nu}^n, \boldsymbol{\theta}^n) \frac{\partial \mathbf{h}_{t-1}^{c1,n}}{\partial \boldsymbol{\nu}^{\log,n-2}} - \boldsymbol{\phi}(\boldsymbol{\nu}^n, \boldsymbol{\theta}^n) \frac{\partial \mathbf{h}_{t-1}^{c2,n}}{\partial \boldsymbol{\nu}^{\log,n-2}} + \boldsymbol{\gamma}^n \odot \mathbf{W}_x^{c1,n}$$

$$\mathbf{f}'(\cdot) \begin{bmatrix} \mathbf{g}(\boldsymbol{\nu}^{n-1}, \boldsymbol{\theta}^{n-1}) \frac{\partial \mathbf{h}_{t-1}^{c1,n-1}}{\partial \boldsymbol{\nu}^{\log,n-2}} - \boldsymbol{\phi}(\boldsymbol{\nu}^{n-1}, \boldsymbol{\theta}^{n-1}) \frac{\partial \mathbf{h}_{t-1}^{c2,n-1}}{\partial \boldsymbol{\nu}^{\log,n-2}} + \boldsymbol{\gamma}^{n-1} \odot \mathbf{W}_x^{c1,n-1} \frac{\partial \mathbf{h}_t^{n-2}}{\partial \boldsymbol{\nu}^{\log,n-2}} \\ \mathbf{g}(\boldsymbol{\nu}^{n-1}, \boldsymbol{\theta}^{n-1}) \frac{\partial \mathbf{h}_{t-1}^{c2,n-1}}{\partial \boldsymbol{\nu}^{\log,n-2}} + \boldsymbol{\phi}(\boldsymbol{\nu}^{n-1}, \boldsymbol{\theta}^{n-1}) \frac{\partial \mathbf{h}_{t-1}^{c1,n-1}}{\partial \boldsymbol{\nu}^{\log,n-2}} + \boldsymbol{\gamma}^{n-1} \odot \mathbf{W}_x^{c2,n-1} \frac{\partial \mathbf{h}_t^{n-2}}{\partial \boldsymbol{\nu}^{\log,n-2}} \end{bmatrix}$$

$$\frac{\partial \mathbf{h}_t^{n-2}}{\partial \boldsymbol{\nu}^{\log,n-2}} \in \mathbf{R}^{2d_{n-2}} \text{ Can be calulated with linear complexity.}$$

$$\frac{\partial \mathbf{h}_{t-1}^{c1,n-1}}{\partial \boldsymbol{\nu}^{\log,n-2}} \in \mathbf{R}^{d_{n-1} \times d_{n-2}} \text{ Saved from previous timestep.}$$

$$\frac{\partial \mathbf{h}_{t-1}^{c2,n-1}}{\partial \boldsymbol{\nu}^{\log,n-2}} \in \mathbf{R}^{d_{n-1} \times d_{n-2}} \text{ Saved from previous timestep.}$$

$$\frac{\partial \mathbf{h}_{t-1}^{c1,n}}{\partial \boldsymbol{\nu}^{\log,n-2}} \in \mathbf{R}^{d_n \times d_{n-2}} \text{ Saved from previous timestep.}$$

$$\frac{\partial \mathbf{h}_{t-1}^{c2,n}}{\partial \boldsymbol{\nu}^{\log,n-2}} \in \mathbf{R}^{d_n \times d_{n-2}} \text{ Saved from previous timestep.}$$

$$\tag{44}$$

Let's define $J_{i,i-1} = \{\frac{\partial \mathbf{h}_{t-1}^{c_1,i}}{\partial \boldsymbol{\nu}^{\log,i-1}}, \frac{\partial \mathbf{h}_{t-1}^{c_2,i}}{\partial \boldsymbol{\nu}^{\log,i-1}}\}$. Then, to calculate the gradient of the hidden units from layer $n$ w.r.t the parameters from layer $n-1$, we need save $J_{n,n-1}$, and to calculate the gradient of the hidden units from layer $n$ w.r.t the parameters of layer $n-2$, we need to save $J_{n,n-2}$ and $J_{n-1,n-2}$. If we keep going, to calculate the gradient of the hidden units of layer $n$ w.r.t the parameters of the first layer, we need to save $J_{n,1}, J_{n-1,1}, J_{n-2,1}, \ldots, J_{2,1}$.

### E.6 Implementing RTRL within the reverse-mode automatic differentiation

For a function $f : \mathbb{R}^n \to \mathbb{R}^m$, we have the Jacobian $\partial f(x) \in \mathbb{R}^{m \times n}$ and we can calculate this Jacobian in two ways: forward-mode or reverse-mode differentiation. In the forward-mode differentiation, the chain rule is applied to each operation while traversing the computational graph in the forward pass [2]. While computing the derivatives during the forward pass is appealing, we need to do $n$ forward passes to get the full Jacobian, where each forward pass would allow us to compute the derivative w.r.t only one of the inputs. i.e., with forward mode differentiation, we evaluate the Jacobian one column at a time. As a result, forward-mode differentiation is inefficient for neural networks; neural networks map from learnable parameters, which can be in millions, to a loss function, hence, have very wide jacobians, $n \gg m$.

Reverse-mode differentiation, on the other hand, offers a more efficient approach. It allows us to evaluate the Jacobian one row at a time, which is particularly advantageous for neural networks. This efficiency comes at the cost of two passes through the network: a forward pass for function evaluation and a backward pass for derivative evaluation [2].

RTRL is an instance of forward-mode differentiation; during the forward pass, the gradient information is evaluated along with the recurrent function computation. As a result, there is no need to perform a backward pass for the recurrent component. To efficiently use a recurrent layer with RTRL within a larger neural network, we combine RTRL for the recurrent layer with the reverse mode for the rest of the network. We use a stop gradient operation on the recurrent layer hidden state and do a normal reverse-mode differentiation. Due to the stop gradient operation, the gradient from the reverse mode assumes no time dependencies between the recurrent states. We then use the gradient traces calculated during the forward pass of the recurrent layer to correct the gradient from the reverse mode and account for the time dependencies between the recurrent states [3]. [4]

## F Additional Details on Trace Conditioning Experiments

In animal learning, *Trace conditioning* is a type of experiment where animals predict the occurrence of a stimulus (e.g., food), based on the occurrence of another stimulus like a tone. There is no prior connection between the two stimuli. However, after enough repetitions of pairing them together—playing the tone and then serving the food—the animal learns to anticipate food arrival when it hears the tone [36]. We use an open-source *trace conditioning* benchmark introduced in prior work [39]. Two signals appear sequentially: the Conditional Stimulus (CS) and the Unconditional Stimulus (US). The CS is the trigger signal, similar to the tone, and the US is the signal of interest and appears several time steps after the CS, similar to the food. The agent also observes several distractor signals which are uncorrelated with the CS and US; the agent must learn ignore them and focus only on predicting the US.

The agent's objective is to predict the onset of the US, which we model as a prediction of the discounted sum of the future US, $G_t$:

$$G_t \doteq \sum_{k=0}^{\infty} \gamma^k \mathrm{US}_{t+k+1}, \tag{45}$$

where $\gamma$ is a discount factor determining the prediction horizon. This problem is challenging because the CS appears, then disappears, and sometime later the US appears; the agent must construct an internal state that represents the time period between the two signals.

---

[4]This can be implemented by defining a custom vjp for the recurrent layer, which modifies the backward pass for the recurrent layer to include the gradient traces.

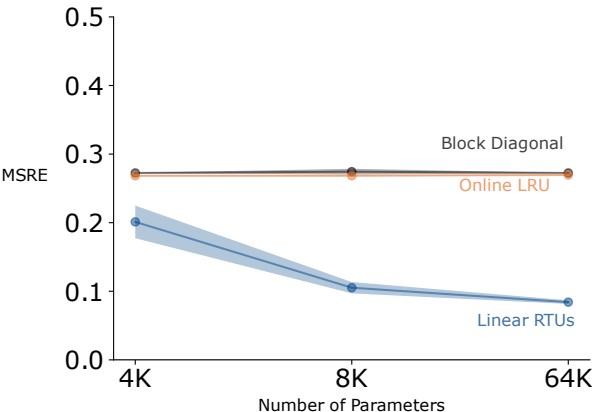

Figure 11: Comparison to a block diagonal RNN.

## F.1 Additional Trace Conditioning Experiments

**Comparison to Other RTRL-based Architectures:** We now compare RTUs to other RTRL-based approaches with similar architectures: an online version of LRU [48] and a vanilla block diagonal RNN. The block diagonal is a recurrent formulation similar to RTU but ignores the relation between the learnable parameters. i.e., replaces $\mathbf{c}_k$ in 3.2 with $\mathbf{c}_k = \begin{bmatrix} a_k & b_k \\ c_k & d_k \end{bmatrix}$.

The results in figure 11 indicate that these seemingly small differences between the diagonal RNNs can result in significantly different behavior. RTUs outperform online LRUs, with the differences discussed in-depth in Section 3.4. RTUs also outperform the block diagonal RNN. We emphasized using real-valued diagonals implicitly assumes symmetric matrices, but that is for a single real-value. This block diagonal has more representational capacity than the RTU. This result suggests it is beneficial for learning to enforce these constraints on the learnable parameters, that they correspond to the rotational representation of complex numbers.

**On the role of RTRL in RTUs:** To highlight the role of RTRL in RTUs, we evaluated the performance of both linear and non-linear RTUs with T-BPTT. In this experiment, we all agents use the same number of parameters; the only difference is whether they use RTRL or BPTT.

Figure 12 summarises the results of this experiment. We can see that the performance of T-BPTT approaches the performance of RTRL as the truncation length increases to cover the whole context of the task.

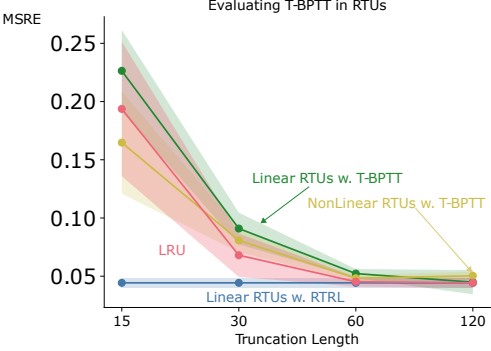

Figure 12: Evaluating RTUs with BPTT.

## F.2 Trace Conditioning Experiments Details

All agents have one recurrent layer, either an RTU or a GRU, and one linear layer. At each time step $t$, the agent passes the observation $\mathbf{o}_t$ to the recurrent layer, which outputs the recurrent state, the agent state. The recurrent state is then passed to the linear layer generating the prediction. For each agent, we swept over the learning rate $\alpha$ used to update the network parameters, $\alpha \in \{10^{-1}, 10^{-2}, 10^{-3}, 10^{-4}, 10^{-5}, 10^{-6}\}$, and averaged the performance for each learning rate over 5 independent runs. We then selected the best-performing learning rate for each agent and ran 30 independent runs using it. For all the experiments, we ran the agents for 2 million steps, and the performance was the mean squared prediction error averaged over the 2 million steps.

## F.3 Learning Rate Sensitivity

We show the learning rate sensitivity for all agents in the animal learning benchmark.

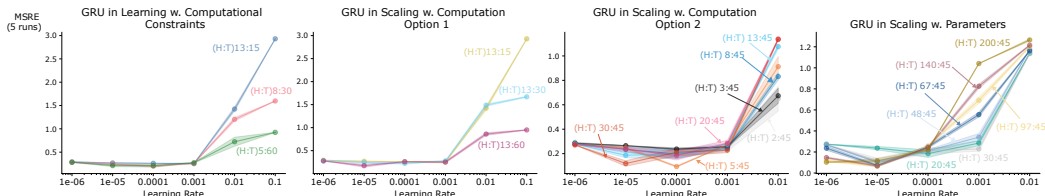

Figure 13: GRUs used in the animal learning benchmark. The *(H: T)* in the label refers to the (hidden dimension: truncation length) for the GRU.

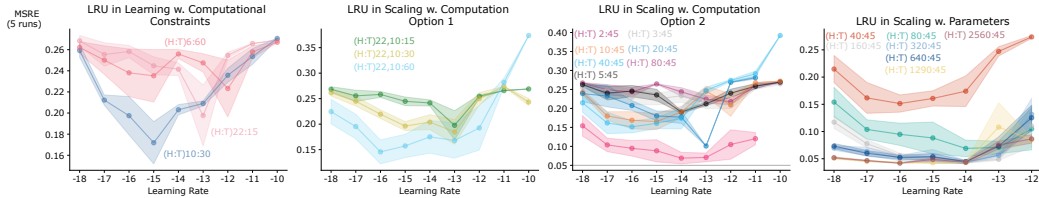

Figure 14: LRUs used in the animal learning benchmark. The *(H: T)* in the label refers to the (hidden dimension: truncation length) for the GRU.

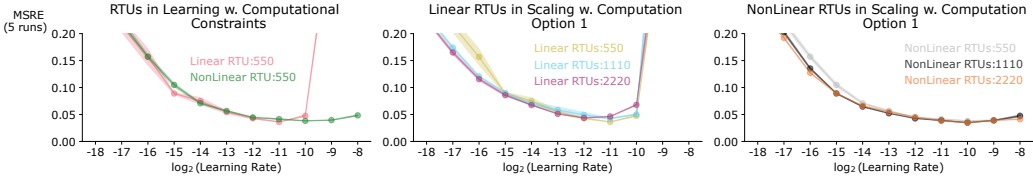

Figure 15: RTUs used in the animal learning benchmark. The number in the label refers to the number of hidden units in the RTU.

## G   Integrating Linear RTRL Methods with PPO

When performing batch updates, as with PPO, the RTRL gradients used to update the recurrent parameters will be stale, as they were calculated during the interaction with the environment w.r.t old policy and value parameters. One solution to mitigate the gradient staleness is to go through the whole trajectory after each epoch update and re-compute the gradient traces. However, this can

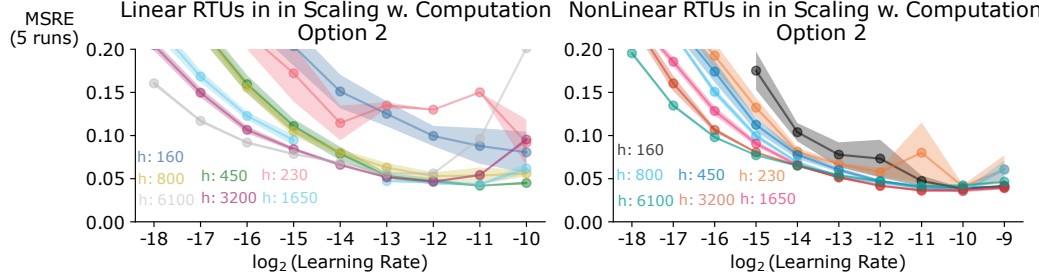

Figure 16: RTUs used in the animal learning benchmark.

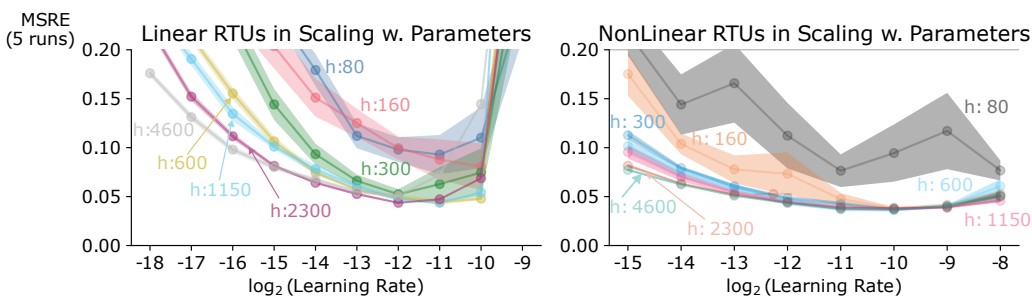

Figure 17: RTUs used in the animal learning benchmark.

be computationally expensive. In Algorithm 1, we provide the pseudocode for integrating RTRL methods with PPO with optional steps for re-running the network to update the RTRL gradient traces, the value targets, and the advantage estimates.

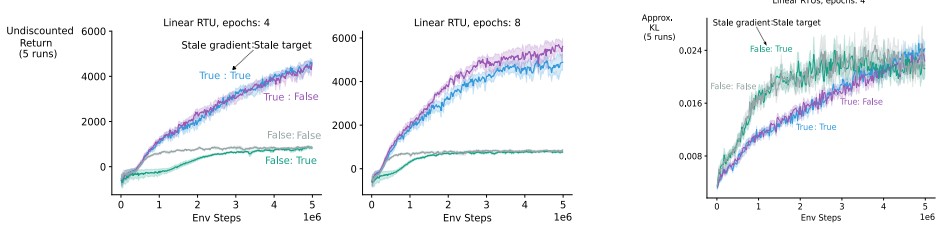

Figure 18: The impact of stale gradients and stale targets when combining RTRL and PPO on Ant.

In the next experiment, we investigate the effect of the gradient staleness in RTRL when combined with PPO and how this staleness interacts with the targets and advantage estimates. To understand this interaction, we evaluate all combinations of stale gradient and stale targets with increasing the number of epoch updates. We perform this analysis on the Ant-P environment from the Mujoco POMDP benchmark [31, 12, 27, 32]. Surprisingly, Figure 18 shows that using a stale gradient results in better performance with RTUs than re-computing the gradient traces. This performance improvement is also consistent when we increase the number of epochs from 4 to 8. It also shows that re-computing the value targets and advantage estimates has a minimal effect on the performance. We repeated the same experiments for NonLinear RTUs and Online LRU with consistent results in figures 19 and 20.

One hypothesis for the superior performance of stale gradients is that the staleness is helping PPO maintain the trust region. We investigate this hypothesis by measuring the KL divergence between the policy used to collect the trajectory and the most recent policy. We use the following estimate for

the KL divergence between the two policies as $(r - 1) - \log(r)$, where $r = (\pi_{\theta_{new}})/(\pi_{\theta_{old}})$. The rightmost subplot of Figure 18 shows that at the beginning of learning, agents with stale gradients move away from the old policy more slowly than agents with fresh gradients; perhaps stale gradients help with maintaining the trust region. However, this hypothesis still needs more investigation in future work.

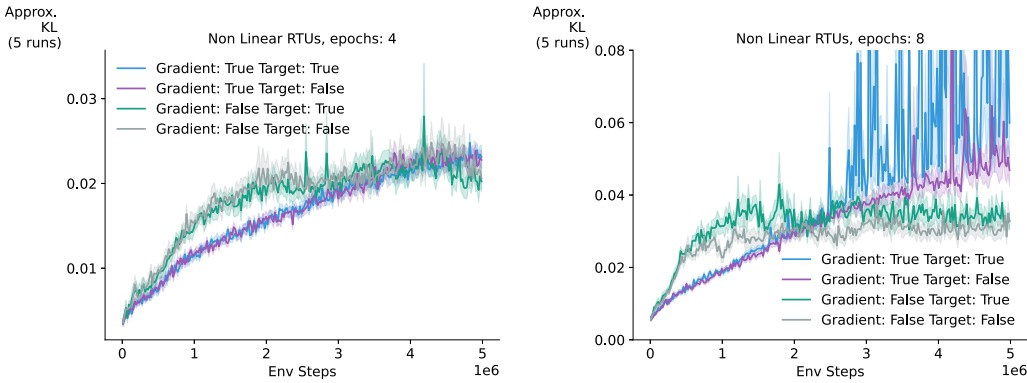

Figure 19: (a) Approximate KL divergence for NonLinear RTU with $4$ epochs.(b) Approximate KL divergence for NonLinear RTU with $8$ epochs.

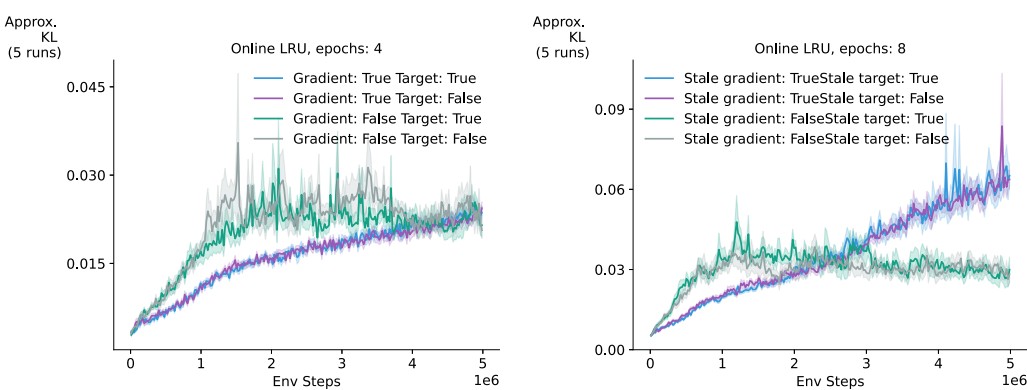

Figure 20: (a) Approximate KL divergence for LRU with $4$ epochs.(b) Approximate KL divergence for LRU with $8$ epochs.

## H    More Details on the Memory-Based Control Experiments

In our implementations, we use a shared representation learning network followed by two MLPs for the actor and the critic heads, as illustrated in Figure 21. The shared representations consist of a feedforward layer with $64$ hidden units and memory components: an RTU, LRU, or a GRU. The actor and the critic's heads consist of two feedforward layers with tanh activation function.

**Additional Mujoco Results:**

In Figures 22 and 23, we set the truncation length for GRU and LRU to be $64$, which is larger than needed to solve the Mujoco POMDP tasks. These results show that even when the truncation length is larger than needed, RTUs still outperform T-BPTT baselines. We also show that the transformer-based models, GPT2, perform worse than RNN-based models. This is consistent with previous work suggesting that transformers might not be suitable for RL tasks [32].

---

**Algorithm 1** Pseudocode for integrating RTRL methods with PPO

---

**Inputs:** a differentiable policy parametrization $\pi(a|s, \mathbf{W}_p)$.
**Inputs:** a differentiable state-value function parametrization $\hat{v}(s, \mathbf{W}_v)$.
**loop**
    Generate a trajectory using the current policy $\mathbf{O}_0, A_0, R_1, \ldots, \mathbf{O}_M, A_M, R_M$,
    Store the transition and the gradient traces for the recurrent components along the way for $i = 0, \ldots, M$
    Compute the advantage estimates and the target value for each timestep $t$
    **for** epoch = 1, ..., k **do**
        Divide the trajectory into minibatches and shuffle them.
        **for** minibatch = 1, ..., m **do**
            Calculate PPO loss
            Perform a gradient step with AutoDiff and correct it with the RTRL saved gradient as discussed in E.6.
        **end for**
        [Optional] Re-run network to update hidden states and the gradient traces for the trajectory.
        [Optional] Update value targets and advantages estimates for the trajectory.
    **end for**
**end loop**

---

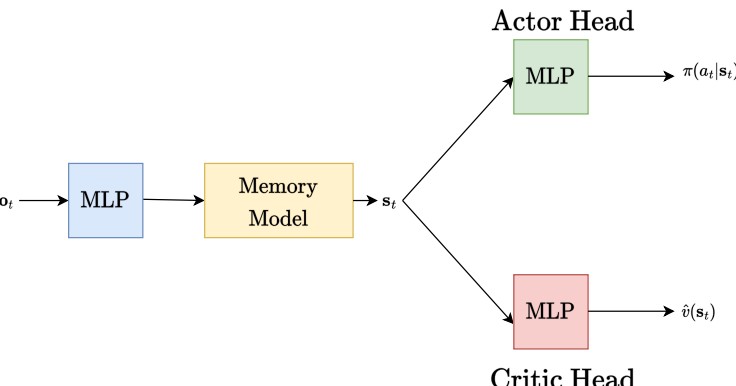

Figure 21: Agents architectures used in our control experiments.

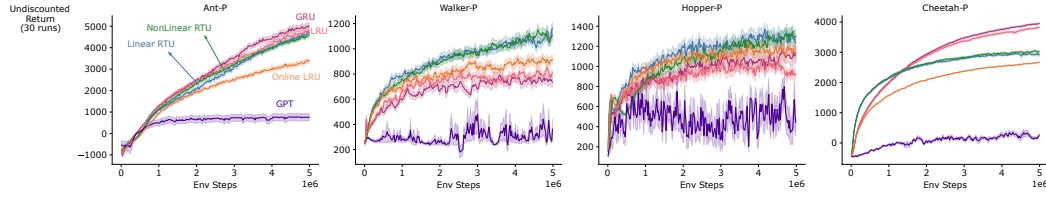

Figure 22: Additional results on Mujoco-P, where we allow GRU and LRU to use a larger truncation length than needed to solve these tasks. We also show results for GPT2.

**Learning Rate Sensitivity:** Figures 24, 25, 26, and 27 show the learning rate sensitivity for all agents in the Mujoco POMDP benchmark. Finally, we used the default hyper-parameters for PPO [40] indicated in Table 1 for all agents.

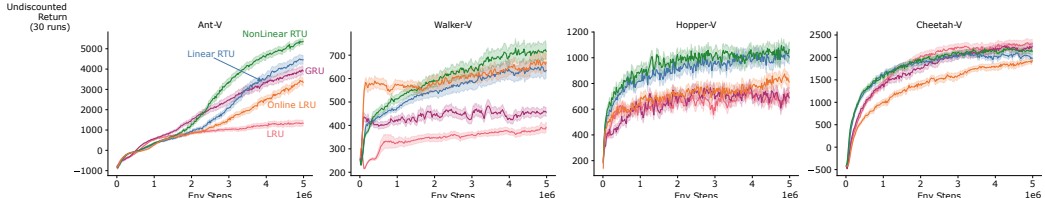

Figure 23: Additional results on Mujoco-P, where we allow GRU and LRU to use a larger truncation length than needed to solve these tasks.

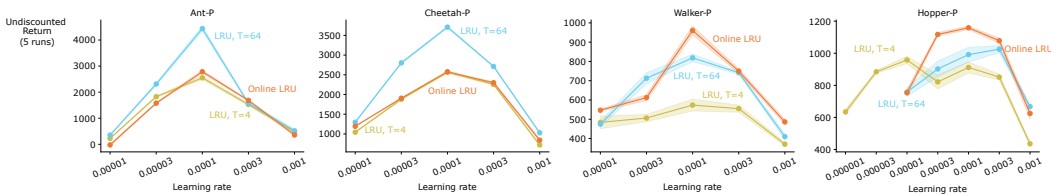

Figure 24: Learning rate sweep for LRU in the control experiments.

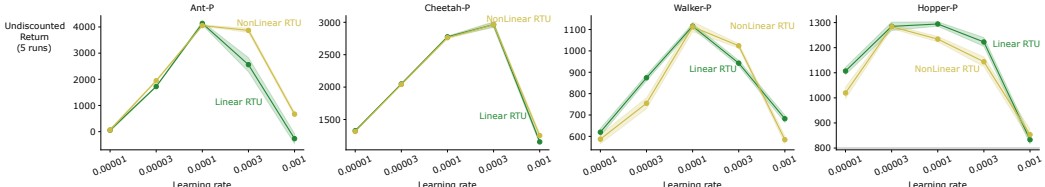

Figure 25: Learning rate sweep for RTUs in the control experiments.

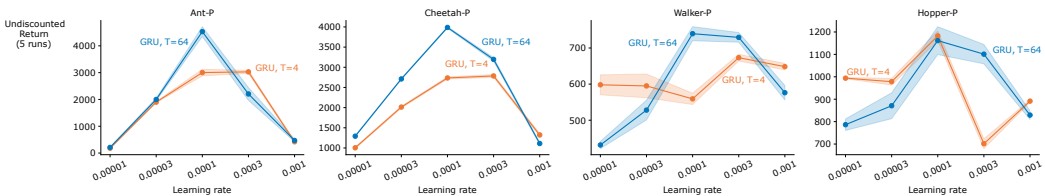

Figure 26: Learning rate sweep for GRUs in the control experiments.

# I   Compute resources

We ran the Mujoco-P, Mujoco-V on NVIDIA P100 GPU. Each of the Mujoco-P and Mujoco-V trials took around 30 minutes to complete on a single GPU. For the POPGym experiments and animal learning experiments, we used a large CPU cluster. Each trial of the POPGym experiments took around 2 hours to complete. While each run of animal learning took around 15 minutes to complete on a single CPU with memory less than 1 GB.

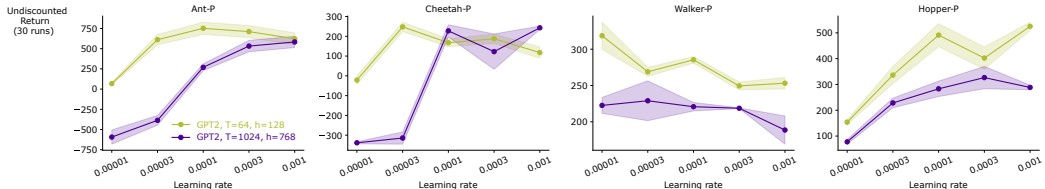

Figure 27: Learning rate sweep for GPT in the control experiments.

| Name | Value |
|---|---|
| Buffer size | 2048 |
| Num epochs | 10 |
| Number of Mini-batches | 32 |
| GAE,$\lambda$ | 0.95 |
| Discount factor, $\gamma$ | 0.99 |
| policy clip parameter | 0.2 |
| Value loss clip parameter | 0.5 |
| Gradient clip parameter | 0.5 |
| Optimizer | Adam |
| Optimizer step size | $[1e-05, 3e-05, 1e-04, 3e-04, 1e-03]$ |

Table 1: Hyper Parameters for PPO.

