# OpenReview forum: "Real-Time Recurrent Learning using Trace Units in Reinforcement Learning"
_NeurIPS.cc/2024/Conference — NeurIPS 2024 poster_

### Official Review · Reviewer_X1j7 · 2024-07-01

**Soundness:** 2
**Presentation:** 3
**Contribution:** 2
**Rating:** 5
**Confidence:** 2

**Summary:**

This paper investigated using complex-valued diagonal RNNs for online RL to provide a small modification (RTUs), and the authors found policy performs significantly better in online RL across various partially observable prediction and control settings.

**Strengths:**

1. The paper is well-written.

2. Method part has good theoretical basis and clear reasoning logic.

3. The experimental results verify the effectiveness of the algorithm in a large number of scenes.

**Weaknesses:**

N/A

**Questions:**

N/A

**Limitations:**

The article discusses the limitations of the method in detail, which I think is reasonable.

---

### Official Review · Reviewer_d6Pv · 2024-07-05

**Soundness:** 3
**Presentation:** 3
**Contribution:** 3
**Rating:** 6
**Confidence:** 4

**Summary:**

This paper introduces Recurrent Trace Units (RTUs), a lightweight extension to linear recurrent units (LRUs) that are more suitable for online reinforcement learning. In a range of ablations and experiments the authors show that RTU trained with Real-Time Recurrent Learning (RTRL) performs on par or outperforms LRU trained with RTRL and GRUs trained with BPTT in online reinforcement learning.

**Strengths:**

- the paper is sound
- the contribution is clear
- extensive ablations on toy experiments
- interesting analysis

**Weaknesses:**

- minor contribution
- some tasks are not very complex (Mujoco-P and Mujoco-V do not require much memory).
- expensive in the multilayer setting, making it limited to simple tasks.

**Questions:**

- in bptt the history is often needed to be truncated because of vanishing/exploding gradients. In theory, RTRL should also suffer from these problems, how are they mitigated here? Is it because of the stale gradients in practice?
- RTUs could be trained with BPTT. This would raise the question whether the increase in performance is exclusively in RTRL.
- It would be interesting to discuss the differnce between the RTRL approach with the approach in [1], where the authors train general stateful policies with stochastic gradient approximation instead of BPTT; especially because the experiments are related.


[1] Time-Efficient Reinforcement Learning with Stochastic Stateful Policies

**Limitations:**

The authors made the main limitation of this method clear. The paper would benefit from answering the above questions.

**Some minor points**:
- line 118: where did the non-linearity f come from (compare left and right)?
- line 148: “Each element of $h_{t-1}$” should be $h_t$
- Why not keeping the $\bar{h}_t$ notation from 3.1 in 3.2?
- the paper would benefit from proofreading; it has a couple of grammatical mistakes like "A window of past observations does not scalable well to long 30 sequences [...]"

**Conclusion**

The paper is nicely structured, with ablations and analysis, and the claimed contribution is fulfilled. Addressing the points above can improve the paper, but the contribution remains small.

---

> ### Author Rebuttal · Authors · 2024-08-06
>
> We thank the reviewer for the detailed and valuable feedback. Here, we respond to the points the reviewer mentioned.
>
> >some tasks are not very complex (Mujoco-P and Mujoco-V do not require much memory).
>
> While Mujoco-P and Mujoco-V could be considered short-term memory tasks, Reacher POMDP and some of the POPGym environments are long-term memory tasks. Table 1 in Ni et al. (2023) provides an estimate of the memory length needed to solve some tasks, and both Reacher-POMDP and Autoencode from POPGym have long memory requirements. Other environments we tested, such as CountRecall, Repeat First, and Concentration, also need a long history to be solved effectively.
>
> [1] Tianwei Ni, Michel Ma, Benjamin Eysenbach, and Pierre-Luc Bacon. When do transformers shine in rl? decoupling memory from credit assignment. In the Thirty-seventh Conference on Neural Information Processing Systems, 2023.
>
> >expensive in the multilayer setting, making it limited to simple tasks.
>
> It’s true that getting the exact gradient for the multi-layer case is expensive. A common solution to this problem is to stop the gradient between the recurrent layers’ hidden states [1][2], this approach has been shown to work in practice and can be used when scaling our RTUs for multi-layers.  While this solution has been shown to work in practice, we think that a more principled approach is still needed.
>
> [1] Kazuki Irie, Anand Gopalakrishnan, and Jürgen Schmidhuber. Exploring the promise and limits of real-time recurrent learning, 2023
>
> [2]Nicolas Zucchet, Robert Meier, Simon Schug, Asier Mujika, and João Sacramento. Online learning of long range dependencies. In Advances in Neural Information Processing Systems, 2023.
>
> >in bptt the history is often needed to be truncated because of vanishing/exploding gradients. In theory, RTRL should also suffer from these problems, how are they mitigated here? Is it because of the stale gradients in practice?
>
> That’s correct, RTRL can still suffer from vanishing/exploding gradients. In our parametrization, we can prevent vanishing/exploding gradient by restricting the magnitude of the complex number to be $\in (0,1]$. We discuss this point first in Appendix D.1, showing that different parametrization requires different restrictions to avoid vanishing/exploding gradients and that we choose the cosine representations as it has the least restrictions. In Appendix C.1, we discuss several ways to enforce the restriction on the magnitude of complex numbers and provide an ablation over these different approaches.
>
>
> >RTUs could be trained with BPTT. This would raise the question whether the increase in performance is exclusively in RTRL.
>
> That’s true. RTUs can be trained with both BPTT and RTRL. We show an experiment in Appendix E.1 (Figure 11) comparing RTUs when trained with BPTT and RTRL. RTRL does play an important role in the improved performance of RTUs over other architectures; RTUs with BPTT will have the same issue of truncation length/computational complexity trade-off. However, the parametrization of RTUs was needed to get to a tractable RTRL.
>
> >It would be interesting to discuss the differnce between the RTRL approach with the approach in [1], where the authors train general stateful policies with stochastic gradient approximation instead of BPTT; especially because the experiments are related.
>
> Thank you for pointing out this paper, we will add it to our discussion on previous work. While the paper introduces a way of calculating an unbiased gradient estimation of BPTT, they note that their estimation has  high variance since they introduce stochasticity to the policy's internal state. This is in contrast to our approach where there is no added stochasticity. Additionally, the proposed approach in [1] was not able to learn both the policy and the value function at the same time without providing additional privileged information to the value function. Our approach doesn’t have these limitations.
>
> >line 118: where did the non-linearity f come from (compare left and right)?
>
> That’s a typo. The f shouldn’t be there at this point of the derivation. Thank you for pointing it out; we will fix it.
>
> > line 148: “Each element of ht−1” should be ht
>
> That’s correct. Thank you for pointing it out.

---

> > ### Comment · Reviewer_d6Pv · 2024-08-12
> >
> > I appreciate the detailed response to my questions. The authors did clarify my concerns, which is why I will raise my score to weak accept.

---

> > > ### Author Response · Authors · 2024-08-12
> > >
> > > Thank you for the response and for updating the score.

---

### Official Review · Reviewer_PH1W · 2024-07-10

**Soundness:** 3
**Presentation:** 3
**Contribution:** 3
**Rating:** 6
**Confidence:** 4

**Summary:**

This work proposes Recurrent Trace Unit (RTU) as a modified variant of the linear recurrent unit (LRU) which has gained some popularity recently as a linear complexity model.

RTU adopts cosine representations of LRU to manipulate two-coupled real-valued hidden states, and introduces non-linearity (while breaking the equivalence to dense linear RNN when non-linearity is introduced in the recurrence).

Like LRU, RTU allows for tractable real-time recurrent learning (RTRL) through diagonal recurrence.
The focus of this work is on evaluating RTU-RTRL for reinforcement learning (RL) in partially observable environments.

The main experiments are conducted on two partially observable variants of six MuJoCo environments for general evaluation; in addition, one MuJoCo environment (Reacher) and five POPGym environments are used as memory tasks. Positive results for RTU are consistently shown.

**Strengths:**

* Evaluating variants of LRU using RTRL *for reinforcement learning* is novel and interesting.

* Beyond the focus on the RL applications, several interesting details about LRU are discussed while developing RTU (Sec 3.2 and 3.4), which should be of interest to people interested in LRU in general.

* Experiments are conducted on several relevant environments.

Overall, I'm supportive of acceptance, provided that the authors will address the main issues described below.

**Weaknesses:**

**One major issue is the repeated claims about outperforming "Transformers" while no such experiment is provided, certainly not in the main text (not even in the appendix as far as I can tell).** See:
- Abstract Line 11, "RTUs significantly outperform GRUs and Transformers"
- Line 329: "We also included a GPT2-transformer baseline"
- Line 330: "We provide the results for GPT2 in Appendix G."
- Conclusion: Line 356: "performed better than the more computationally intensive GRUs and Transformers"

I could not find the Transformer/GPT2 results even in the indicated Appendix G, Figures 21 or 22 either.

Given all these emphases on this claim, I had expected to see such results in the main text, e.g., in the same style as Figure 4 and 6.

I initially considered putting 1 ("incorrect statements") to encourage the authors to fix this immediately, but in the end, I decided to put the score I would give disregarding any mentions to Transformers. Regarding Transformers, please also check the related comment in the "Questions" field below.

**There are also some clarity issues:**

* Clearer explanations to explain the gap between RTU and LRU are expected.

In Figure 1, some explanations are needed to help the readers understand why there is such a big gap between LRU and RTU when everything is equal except the architecture. Is there anything specific to RTRL? or do you observe similar trends when T-BPTT is used for both models?

Similarly, some explanations should be provided to explain why Online LTU largely underperform RTU models in Figures 4, 5, and 6.

The only related comment I could find is Line 209: *"We found small choices in our implementation for LRU did not always behave as expected, partially due to how auto-differentiation is implemented in packages for complex numbers"*
So is LRU's problem just an implementation issue? Please clarify.

* I find Figure 2 misleading as they mix model architectures and learning algorithms within the same comparison, i.e., they compare LRU-TBPTT with RTU-RTRL.

Instead, the comparison should be between LRU-TBPTT vs. RTU-TBPTT vs. GRU-TBPTT, and LRU-RTRL vs. RTU-RTRL separately. The statement *"LRU and GRU with T-TBTT is not competitive with RTUs"* (caption Figure 2) is true but does not allow us to conclude on the superiority of RTU over LRU since two different learning algorithms are used. I acknowledge that good results are sufficiently shown later in Figures 4, 5 and 6, but Figure 2 alone is not informative.

**Some of the experimental designs are not convincing:**

* (Related to the point above) I find the experimental setting of Sec 4.2/Figure 2 under "resources constraints" too artificial (therefore not particularly useful). Does the chosen "computational budget of 15000 FLOPs" (Line 237) correspond to something intuitive/useful? How does the memory requirement differ between RTU-RTRL and RTU-TBPTT for different values of T?

**Questions:**

Most of the questions have already been asked above. These are comments and suggestions.

* Tractable RTRL with a diagonal recurrent matrix dates back to [R1] [R2] at least, which should be cited.

[R1] Gori et al. IJCNN 1989. BPS: A learning algorithm for capturing the dynamic nature of speech.

[R2] Mozer. Complex Systems 1989.  A focused backpropagation algorithm for temporal pattern recognition.

* The authors' view on Transformers for POMDPs (the second paragraph of the introduction; starting at Line 20) is restricted as it lacks discussion about the "linear" variants of Transformers that are stateful and permit a "recursive" formula just like RNNs.
There are several prior works evaluating [R3] [R4] [R5] and discussing [R6] such models in the context of reinforcement learning.

[R3] Irie et al. NeurIPS 2021. Going Beyond Linear Transformers with Recurrent Fast Weight Programmers.

[R4] Irie et al. ICML 2022. A Modern Self-Referential Weight Matrix That Learns to Modify Itself.

[R5] Pramanik et al. arXiv 2023. Recurrent Linear Transformers.

[R6] Lu et al. ICML 2024. Rethinking Transformers in Solving POMDPs.

* In the equation just below Line 118 (sec 3.1), *"f("* is a typo at this stage (the non-linearity is not introduced yet).

* Regarding the non-linearity "f", for the tractable RTRL to hold, "f" has to be a *purely* element-wise function. For example, softmax or layernorm would not work there (as they introduce interactions between recurrent units); I know nobody would use softmax or layernorm as a recurrent non-linearity in practice, but it might make sense to point out that there is a condition on "f", in the strict mathematical sense.

* It is unfortunate that no further architectural advances have been integrated and evaluated. Based on [11], gating can be made compatible with RTRL, and a recent model that is closely related to LRU, "Mamba", also puts back gating to linear recurrence. So some gating could have been a natural extension too.

* The proposed RTU is not specific to RL. I'm wondering if the authors considered applying it to other supervised learning applications.

**Limitations:**

Comments are already provided above.

---

> ### Author Rebuttal · Authors · 2024-08-06
>
> We thank the reviewer for the detailed and valuable feedback. Here, we respond to the points the reviewer mentioned.
>
> > One major issue is the repeated claims about outperforming "Transformers" while no such experiment is provided.
>
> We do provide the results for GPT-2 in Appendix G. Figure 21 shows the GPT-2 performance on Mujoco-P tasks, and Figure 26 has the learning rate sweep results for GPT-2. In Figure 21, the GPT-2 line is far below all the baselines we have, so it’s easy to miss. We changed it to a brighter color now, so it’s obvious. Please check the uploaded PDF for the updated figures.
>
> The reason for not including GPT-2 in the main text is that the number of parameters in GPT-2 is way more than any of the architectures we have, so the comparison is a bit unfair. We will modify the abstract so as to not put a lot of emphasis on this comparison since it’s not a core result of the paper.
>
> >In Figure 1, some explanations are needed to help the readers understand why there is such a big gap between LRU and RTU
>
> Our goal from Figure 1 was twofold:
>  1. Understand the utility of RTU parametrization in contrast to LRU.
>  2. Understand the effect of different design choices, such as non-linearity and using a linear projection layer.
> By controlling for other factors, we can ascertain that the gap in performance is due to the difference in parametrization choices, i.e., using explicit complex numbers in LRU versus using rotation matrices in RTUs. Using rotation matrices avoids the complications around taking the real part of the hidden states; we show in Appendix D.2 that taking the real part of the hidden states could result in a biased gradient estimate.
>
> For Figure 1, we chose to use RTRL as using T-BPTT would complicate the comparison. Bad performance could be attributed to the truncation length rather than the architecture itself, while with RTRL, there is no confounding factor other than the parametrization.
>
> >Similarly, some explanations should be provided to explain why Online LTU largely underperform RTU models
>
> When comparing RTU with Online LRU, the core difference is RTU’s parametrization of complex numbers as rotation matrices. As we noted above, using rotation matrices avoids the complications around taking the real part of the output.  We also noticed that using complex numbers combined with auto-diff libraries resulted in unexpected results when taking the gradients. This appears to be an issue related to the use of complex numbers in general, not specific to LRU. We pointed to a similar discussion in the footnote on page 5. We also emailed Online LRU authors to verify that our implementation is correct, and we used their suggested implementation of Online LRU in our experiments.
>
> >I find Figure 2 misleading as they mix model architectures and learning algorithms within the same comparison.
>
> The goal of Figure 2 was to highlight the trade-off between computational resources and performance when using T-BPTT versus RTRL. So, we could compare RTU-RTRL and RTU-TPBTT. However, we believe the reader would be more interested in seeing the performance of LRU-TBPTT and GRU-TBPTT to see how much benefit we get on top of these more widely used approaches when moving to RTU-RTRL. The plot becomes unreadable with too many choices. But your point is a good one, and we do provide a comparison of RTU-BPTT with LRU BPTT in Appendix E, Figure 11. As expected, when using T-BPTT with RTU, the truncation length affects the performance of RTU as well.
>
> >Does the chosen "computational budget of 15000 FLOPs" (Line 237) correspond to something intuitive/useful?
>
> We wanted to start with a computational budget under which all architectures can learn and produce relatively good performance so the comparison is meaningful. We looked at the sizes of the GRU architectures used in [1], where the animal learning experiments were introduced, and chose similar hidden dimensions to start with. That’s how we came to the choice of 15k flops specifically.
>
> [1] Banafsheh Rafiee, Zaheer Abbas, Sina Ghiassian, Raksha Kumaraswamy, Richard S Sutton, Elliot A Ludvig, and Adam White. From eye-blinks to state construction: Diagnostic benchmarks for online representation learning. Adaptive Behavior, 2020.
>
> >diagonal recurrent matrix dates back to [R1] [R2]
>
> Thanks for pointing out these references. We will cite them in the introduction.
>
> > The authors' view on Transformers is restricted
>
> Thanks for pointing out those relevant papers. We will modify the discussion on transformers to include them. While the approaches in [R3] [R4][R5] are all interesting and relevant, they face the same challenges as T-BPTT-based approaches in terms of the trade-off between sequence length, computational complexity, and performance.
>
>
> >Regarding the non-linearity "f", for the tractable RTRL to hold, "f" has to be a purely element-wise function.
>
> We agree that the use of layer norm as an activation function is not conventional. The discussion we provided in Appendix A.2 was to mathematically state what conditions we need on f for the equivalence to hold. However, as we point out in the main body, we think of these theoretical results as negative results; as you pointed out, those functions are not element-wise and not conventional activation functions.
>
> > Based on [11], gating can be made compatible with RTRL. So some gating could have been a natural extension too.
>
> Thank you for the suggestion. We agree that gating is a natural extension of this work.
>
>  >The proposed RTU is not specific to RL.
>
> We agree that RTUs are not specific to RL. However, we think RTRL approaches are more needed in the context of online RL than supervised learning. As in supervised learning, offline datasets are usually available, and having access to a long history is not as hard as in online RL. There are areas of online supervised learning where RTRL approaches could be useful, which could be explored in future work.

---

> > ### Comment · Reviewer_PH1W · 2024-08-10
> > **Thank you for the response**
> >
> > I thank the authors for their response.
> >
> > > We do provide the results for GPT-2 in Appendix G. Figure 21
> >
> > Thank you for the pointer to Figure 21 and 26, and the updated figures. The authors' comment regarding the "bad" colors in the old figure is accurate; the new colors help.
> >
> > But I also do confirm that it does not make sense to emphasize on Transformer-related claims that are entirely based on results that are hidden in Appendix G, Figure 21 and 26 (and for other reasons I explain below).
> >
> > > The reason for not including GPT-2 in the main text is that the number of parameters in GPT-2 is way more than any of the architectures we have, so the comparison is a bit unfair.
> >
> > It is unreasonable to complain about the model's parameter count, as it is the deliberate choice made by the authors.
> > On the contrary, the current comparison seems unfair to the Transformers.
> > GPT-2 is a language model instantiation of Transformers.
> > To properly evaluate Transformers in specific RL tasks, their hyper-parameters have to be adjusted accordingly. In particular for POMDPs, a discussion on the context length is a minimum requirement in a proper evaluation, since if their context length is shorter than the memory span required for the task, they will obviously fail (we won't even need to run experiments to figure that out).
> >
> > In this sense, I find all the current “GPT-2” related results reported in this paper insignificant, so are the claims regarding the "superiority" over "Transformers".  I understood that this choice of Transformer architecture/GPT-2 was based on a prior work, but I do not see any good reason to follow such a choice which does not make sense.
> >
> > I will maintain my currently score on the condition that the authors promise to remove all these claims from the main text, as I initially stated. Alternatively, the authors should conduct a new set of proper experiments by tuning the Transformer hyper-parameters/architecture.
> >
> > What follows are minor comments:
> >
> > > Our goal from Figure 1 was twofold: Understand the utility of RTU parametrization in contrast to LRU.
> >
> > This was actually my original critique: Figure 1 does not fulfill this goal. Figure 1 currently *shows* the utility of RTU but fails at helping us "understand" why it is better than LRU. It will make sense to add a sentence toward the end of Sec 4.1 to explain this gap and guide readers to take a look at Sec 3.4 and Appendix D.2.
> >
> > > For Figure 1, we chose to use RTRL as using T-BPTT would complicate the comparison. Bad performance could be attributed to the truncation length rather than the architecture itself,
> >
> > I do not agree with this argument. I asked whether the gap between RTU and LRU would remain the same when T-BPTT was used. If we train both models with T-BPTT, the condition is the same for the two models, so the author’s argument does not hold. That said, I had also overlooked some details in Sec 3.4 and Appendix D.2, so I no longer find this experiments crucial.

---

> > > ### Author Response · Authors · 2024-08-11
> > >
> > > We thank the reviewer for the detailed response and feedback.
> > >
> > > We agree that the GPT-2 results are not core to the paper, and we will modify the main text not to emphasize those results. We also wanted to point out that we tuned the context length and the learning rate for GPT-2. In Figure 26 (or Figure 2 in the uploaded pdf), we show two variants of GPT-2, with context lengths of 1024 and 64. Both of these variants should have enough context length for the Mujoco-p benchmark. Nevertheless, we agree with the reviewer, and we will change the main text accordingly.
> > >
> > > > It will make sense to add a sentence toward the end of Sec 4.1 to explain this gap and guide readers to take a look at Sec 3.4 and Appendix D.2.
> > >
> > > Thank you for the suggestion. We will add a couple of sentences to clarify this part.

---

> > > > ### Comment · Reviewer_PH1W · 2024-08-12
> > > >
> > > > > tuned the context length and the learning rate for GPT-2.
> > > >
> > > > Thank you. I had overlooked this too as I could not find any description regarding this in any part of the text.
> > > >
> > > > But then this is one more reason not to call those models GPT-2. I am not aware of any GPT-2 architectures with a context length size of 64. I do not understand what makes this model a GPT-2.
> > > >
> > > > Overall, given that there are prior works successfully applying Transformers to RL settings (e.g., Parisotto et al. [25]); I would not find these Transformer experiments very informative, unless there were more in-depth studies.
> > > >
> > > > I acknowledge the author's promise to modify the corresponding claims. Please ensure these modifications are implemented.

---

### Official Review · Reviewer_EcqN · 2024-07-12

**Soundness:** 3
**Presentation:** 3
**Contribution:** 3
**Rating:** 7
**Confidence:** 4

**Summary:**

This paper introduces a novel complex-valued, diagonally connected recurrent network model based on LRUs called Recurrent Trace Units. Due to the diagonal connectivity, the model can be trained using real-time recurrent learning in linear time. The authors further propose a version of PPO that uses stale gradients computed by RTRL during interaction. The approach is evaluated on a simple online-prediciton task and a range of POMDP RL tasks.

**Strengths:**

The paper is well written overall. All the necessary background information is summarized in a clear fashion. The main idea is laid out concisely. The experiments used for evaluation are adequate and results are displayed in a visually appealing way. Finally, the direction of research is undoubtedly very significant.

**Weaknesses:**

A thorough ablation study of the proposed architecture is missing, i.e. taking discrete steps from LRUs towards RTUs and comparing all of them in one plot, including BPTT.

The work is not overly original, but rather an unavoidable next step after the success of LRUs.

The claim that taking the Real-part of the latent state leads to a biased gradient is not really convincing, all I see is an expression that is harder to compute.

There are many typos and misplaced words.

Somehow, the Appendix was more interesting that the actual paper.

Minor issues:
- L 29: "... does not scalable well ..."
- L 32: "... is well suited for update the state online ..."
- L 48: The sentence starting on this line looks like it needs a citation, please add something like "as we will show in section..."
- L 143, missing word: diagonal "matrix"
- L 151: "... choices made in LRUs, that they showed ..."
- L 291: The sentence starting on this line is confusing.

**Questions:**

- Can you comment on the claim of a biased gradient? Surely, information is lost by discarding the real part, but the gradient should not be *biased*?

- One major advantage of LRUs is the possibility to employ parallel scans reducing the complexity to sublinear in the number of units. Is this advantage thwarted by adding a non-linearity?

**Limitations:**

The authors comment on the limitation regarding multi-layer architectures. Societal impacts of the work are not considerably different than those of Artificial Intelligence in general ...

---

> ### Author Rebuttal · Authors · 2024-08-06
>
> We thank the reviewer for the detailed and valuable feedback. Here, we respond to the points the reviewer mentioned.
>
> >A thorough ablation study of the proposed architecture is missing, i.e. taking discrete steps from LRUs towards RTUs and comparing all of them in one plot, including BPTT.
>
> Thank you for raising this point. Currently, in Figure 1, we provide an ablation on several architectural choices and test them one at a time while fixing everything else. This means that the only difference between RTU and LRU lines in Figure 1 is the RTU parametrization, which is the main difference between LRU and RTU.
>
> In the left plot of Figure 1 (titled: linear recurrence), we are testing the minimal architectural choice that could be made, a linear recurrence. Hence, this plot is only testing the utility of the RTU parametrization over LRU’s with the absence of any other factor. We then move to test different architectural choices that could be made for either of them, such as adding non-linearity or a projection layer.
>
> For Figure 1, we chose to use RTRL as the learning algorithm rather than T-BPTT. We made this choice to prevent any other confounding factor (such as truncation length in T-BPTT) from affecting the performance of either architecture. We still tested RTUs with T-BPTT in the appendix, Figure 11, but as expected with T-BPTT, the performance of RTUs would reduce due to truncated gradients.
>
> >Somehow, the Appendix was more interesting that the actual paper.
>
> We are glad that you found the appendix interesting. We tried to extensively answer any questions that might arise about our approach, but we still didn’t want to overwhelm the reader with a lot of mathematical details in the main body and rather focus on the core ideas, so we chose to delegate most of the mathematical work to the appendix.
>
> >Can you comment on the claim of a biased gradient? Surely, information is lost by discarding the real part, but the gradient should not be biased?
>
> The discarded information from taking the real part also results in discarded gradient components. This makes the gradient estimation different from the true gradient and biased towards the information coming from the real components. We provide a simple derivation of that in Appendix D.2. It is worth noting, though, that the biased gradient issue happens when the recurrent part of LRU is not followed by a linear projection. Hence, for LRU to have an unbiased gradient estimate, it needs to always have a linear projection layer afterward, which is restrictive.
>
> Additionally, while the original LRU paper proposes always having a linear projection after LRU, our empirical evaluations in section 4.1 show that this linearity restriction actually harms LRU's performance.
>
> >One major advantage of LRUs is the possibility to employ parallel scans reducing the complexity to sublinear in the number of units. Is this advantage thwarted by adding a non-linearity?
>
> Currently, Linear RTUs in Eq.2, where we apply non-linearity afterward, not on the recurrence, can still benefit from parallel scans, but the Non-Linear RTUs cannot since the recurrence part is non-linear. However, there are some recent works on parallelizing non-linear recurrence, which could be explored with Non-Linear RTUs as well [1][2].
>
> [1] Lim, Y. H., Zhu, Q., Selfridge, J., & Kasim, M. F. Parallelizing non-linear sequential models over the sequence length. In The Twelfth International Conference on Learning Representations.
>
> [2] Gonzalez, X., Warrington, A., Smith, J. T., & Linderman, S. W. (2024). Towards Scalable and Stable Parallelization of Nonlinear RNNs. arXiv preprint arXiv:2407.19115.

---

> > ### Comment · Reviewer_EcqN · 2024-08-10
> >
> > Thank you for the response and for pointing to some literature on parallelizing non-linear RNNs.
> >
> > > We still tested RTUs with T-BPTT in the appendix, Figure 11, but as expected with T-BPTT, the performance of RTUs would reduce due to truncated gradients.
> >
> > I feel like showing that the model performance increases significantly when trained using RTRL over when trained with BPTT is the main contribution of this paper. Paired with the timing comparison of RTRL and TBPTT over different truncation horizons, this would make a great point and I lament that it did not make it into the main part of the manuscript.
> >
> > > This makes the gradient estimation different from the true gradient and biased towards the information coming from the real components.
> >
> > I am still not convinzed. When only using the real part I expect my gradients to be _biased_ towards the real part because this is the only quantity I am considering after all. Whereas, when using both the real and the imaginary part, I also expect the gradients to contain some portion corresponding to the imaginary part. Comparing the two equations in Appendix D2 seems to be misguided.
> >
> > Nonetheless, I still feel comfortable with keeping my current rating.

---

> > > ### Author Response · Authors · 2024-08-12
> > >
> > > Thank you for the response and feedback.
> > >
> > > > When only using the real part I expect my gradients to be biased towards the real part because this is the only quantity I am considering after all.
> > >
> > > That is correct. We were unclear in our initial response, and we meant that though a reader *might* think we can just take the gradient of the real part, this would not be the same as the actual gradient, and this is what we meant by bias. We realize this word is really not the right word, and we should simply say it would be incorrect to do that to get the original behavior, as taking the real part discards some information.

---

### Official Review · Reviewer_nsAS · 2024-07-13

**Soundness:** 3
**Presentation:** 3
**Contribution:** 3
**Rating:** 6
**Confidence:** 2

**Summary:**

In online reinforcement learning, Recurrent Neural Networks (RNN) are still better than transformers, but there are still problems that need to be addressed. In this paper, the authors propose a modified form of the recurrence equation used for Linear Recurrent Units (LRU) – a type of RNN – called a Recurrent Trace Unit (RTU). The main differences RTU introduces compared to LRU is that it uses a diagonal matrix in place of a full matrix, thus reducing computations and it uses complex valued numbers in the diagonal. Note that multiplications with complex values can be represented as multiplying by a 2x2 matrix with real valued numbers. Additionally, RTUs are a more generalized form of LRUs since they allow nonlinearity in the recurrence equation, unlike LRUs which are strictly linear. This leads to improved accuracy. The authors first explain their recurrent formulation in detail and justify their design choices.

In the second half of the paper the authors empirically perform multiple comprehensive tests that show that RTUs outperform other architectures (GRUs, LRUs, and Transformers [note: Transformer results included in supplementary] ) in online reinforcement learning tasks, providing better performance with less computation.  These tests include ablation, scaling of learning under various constraints, comparison of two different training methods – Real-Time Recurrent Learning (RTRL) (i.e. incremental updates) and Truncated Backpropagation Through Time (T-BPTT) (i.e. batched updates), and analysis of how well RTUs ‘remember’ information compared to other architectures.

**Strengths:**

•	The paper addresses an important problem in online reinforcement learning and proposes a computationally more efficient solution, RTUs, which outperform other architectures in various online reinforcement settings
•	Has comprehensive tests covering and analyzing a variety of cases, which empirically show the improvements from the proposed architecture, RTU.
•	Provides detailed explanations and derivations of the RTUs formulas, making it easier for readers to understand and replicate it
•	Results are explained well using figures
•	Supplementary section goes into detail on many areas which are not fully explained in the main text and answers many questions that may arise.

**Weaknesses:**

•	Most of the comprehensive tests compare the proposed RTU only with GRU and LRU architectures (there are a few transformer (GPT) tests in the supplementary). It would be better if the author uses more architectures or explains why these other two architectures are sufficient for a comparison
•	Explaining the importance and benefits of addressing problems with online reinforcement learning in partially observable environments would strengthen the motivation.
•	Some terminology not completely clear, such as difference between “Reinforcement learning” and “Online reinforcement learning”

**Questions:**

However, is comparing with GRU and LRU enough for online reinforcement learning tasks? Or, would testing different architectures or explaining why the chosen architectures are sufficient would be better?

**Limitations:**

Paper is very comprehensive; it motivates and both mathematically (in the supplementary) and empirically shows the improvement of their proposed architecture very well. One downside is that the paper has a lot of mathematical formulations that may be difficult to follow for someone not well versed in this topic. However, these formulations are necessary and each step of the formulations is explained in detail, making it easier to follow.

The author performs many types of tests comparing the proposed architecture with GRU and LRU, including a few transformer tests in the supplementary. However, is comparing with GRU and LRU enough for online reinforcement learning tasks? Testing different architectures or briefly explaining why the chosen architectures are sufficient would be better.

In the motivation, the author motivates why RNNs are still state of the art in reinforcement learning in partially observable environments and why transformers are not suitable for this task. However, the reason why reinforcement learning in partially observable environments is important is not explained. Having a few sentences on the importance addressing of this topic would strengthen the motivation.

Lastly, in the introduction and abstract, the author mentions both reinforcement learning and online reinforcement learning, but does not explain the differences. The authors should clarify whether these are the same concepts or are distinct.
Grammar:
•	Line 188-190: “RTUs can be 189 seen as a small generalization of LRUs, moving away from strict nonlinearity” shouldn’t this be linearity, since LRU use linear operations?

---

> ### Author Rebuttal · Authors · 2024-08-06
>
> We thank the reviewer for the detailed and valuable feedback. Here, we respond to the points the reviewer mentioned.
>
> > Explaining the importance and benefits of addressing problems with online reinforcement learning in partially observable environments would strengthen the motivation.
>
> We motivated the importance of learning under partial observability in the first paragraph of the introduction when talking about how in the real world, agents perceive their environments through imperfect (i.e., partial) sensory observations. Hence, for agents to be deployed and learn online in the real world, they need the ability to predict and control under partial observability. We agree that the motivation for learning under partial observability is core to our work, and we will expand our current motivation in the introduction to emphasize the importance of learning under partial observability.
>
> >Some terminology not completely clear, such as difference between “Reinforcement learning” and “Online reinforcement learning”
>
> Thanks for pointing that out. When we say "online reinforcement learning" we mean that the agent learns while interacting with the environment. This contrasts with offline reinforcement learning or reinforcement learning with access to a simulator. We will add a couple of sentences to the intro to make this clear.
>
> >is comparing with GRU and LRU enough for online reinforcement learning tasks? Or, would testing different architectures or explaining why the chosen architectures are sufficient would be better?
>
> We chose GRU as our main baseline since it’s widely used for reinforcement learning problems with partial observability and has been shown to outperform most of the other memory components in several domains, which makes it a strong baseline to compare against  [1][2][3][4]. Additionally, the authors of the original POPGym paper that we use in section 6 have extensively tested various recurrent and transformer architectures on the benchmark, and GRU was always the best-performing architecture [1]. Since we are reusing their benchmark and similar other tasks, it is natural to compare against GRU as they have already done extensive testing with the most known architectures.
> While LRU has not been explored in reinforcement learning domains, we added it to our experiments because it is similar to our proposed architecture.
>
>
> [1] Steven Morad, Ryan Kortvelesy, Matteo Bettini, Stephan Liwicki, and Amanda Prorok. POPGym:Benchmarking partially observable reinforcement learning. In International Conference on Learning Representations, 2023
>
> [2] Tianwei Ni, Benjamin Eysenbach, and Ruslan Salakhutdinov. Recurrent model-free rl can be a strong baseline for many pomdps. In International Conference on Machine Learning, 2022.
>
> [3] Danijar Hafner, Jurgis Pasukonis, Jimmy Ba, and Timothy Lillicrap. Mastering diverse domains through
> world models. arXiv preprint arXiv:2301.04104, 2023.
>
> [4] Pleines, M., Pallasch, M., Zimmer, F., & Preuss, M. (2023). Memory gym: Partially observable challenges to memory-based agents. In The eleventh international conference on learning representations.
>
> > One downside is that the paper has a lot of mathematical formulations that may be difficult to follow for someone not well versed in this topic. However, these formulations are necessary and each step of the formulations is explained in detail, making it easier to follow.
>
> We are glad that you found our mathematical formulations easy to follow. We agree that mathematical formulations can be dense, but we think they were all needed to provide the reader with a thorough understanding of our approach and answer any question that might arise about it. We chose to delegate most of the mathematical analysis to the appendix to allow the reader to understand the main contributions easily and then dive into more details about the specifics in the appendix.
>
> >• Line 188-190: “RTUs can be 189 seen as a small generalization of LRUs, moving away from strict nonlinearity” shouldn’t this be linearity, since LRU use linear operations?
>
> That’s correct. The sentence should be “RTUs can be seen as a small generalization of LRUs, moving away from strict linearity”

---

### Author Rebuttal · Authors · 2024-08-06

We would like to thank all of the reviewers for their valuable feedback on the paper. We’ve carefully considered each concern and suggestion and provided detailed responses.
While the reviewers had multiple concerns, there was no major common issue. To further address some of the reviewers' concerns regarding clarity, we added an updated version of Figures 21 and 26, changing the color of the GPT-2 baseline to be more obvious.

---

### Decision · Program_Chairs · 2024-09-25

**Decision:**

Accept (poster)

**Comment:**

Overall the reviewers agreed that the paper makes a nice contribution to deep architectures for reinforcement learning. Given the success of GRUs in the deep RL literature it is impressive to see this level of improvement.

The reviewers noted a number of issues that can use more clarification in the final paper and the authors should work hard to address them.